# Connexin-43-dependent ATP release mediates macrophage activation during sepsis

Michel Dosch, Joël Zindel, Fadi Jebbawi, Nicolas Melin, Daniel Sanchez-Taltavull, Deborah Stroka, Daniel Candinas, Guido Beldi*

Department for BioMedical Research, University of Bern, Bern, Switzerland

**Abstract** Bacterial spillage into a sterile environment following intestinal hollow-organ perforation leads to peritonitis and fulminant sepsis. Outcome of sepsis critically depends on macrophage activation by extracellular ATP-release and associated autocrine signalling via purinergic receptors. ATP-release mechanisms, however, are poorly understood. Here, we show that TLR-2 and −4 agonists trigger ATP-release via Connexin-43 hemichannels in macrophages leading to poor sepsis survival. In humans, Connexin-43 was upregulated on macrophages isolated from the peritoneal cavity in patients with peritonitis but not in healthy controls. Using a murine peritonitis/sepsis model, we identified increased Connexin-43 expression in peritoneal and hepatic macrophages. Conditional $Lyz2^{cre/cre}Gja1^{flox/flox}$ mice were developed to specifically assess Connexin-43 impact in macrophages. Both macrophage-specific Connexin-43 deletion and pharmacological Connexin-43 blockade were associated with reduced cytokine secretion by macrophages in response to LPS and CLP, ultimately resulting in increased survival. In conclusion, inhibition of autocrine Connexin-43-dependent ATP signalling on macrophages improves sepsis outcome.

DOI: https://doi.org/10.7554/eLife.42670.001

*For correspondence:
guido.beldi@insel.ch

## Introduction

Sepsis is associated with high mortality and was now recognised as a Global Health Priority by the World Health Organization (*Reinhart et al., 2017*). Mechanistically sepsis is characterised by a dysregulation of the host immune response to bacteria resulting in local and systemic exacerbated pro-inflammatory response and a concomitant protracted anti-inflammatory response (*Delano and Ward, 2016*; *Hotchkiss et al., 2013*). Accordingly, immuno-regulatory drugs have been considered as a potential treatment, however, clinical results have been unsatisfactory so far (*Remick, 2003*). Despite the urgency to improve patient outcomes and major advances in the understanding of sepsis, a specific therapy is still lacking.

Peritoneal sepsis, as a result of a hollow-organ perforation, is typically associated with invasion of the peritoneal cavity by gut contents containing intestinal bacteria. Intestinal commensals and pathogens proliferate then in the peritoneal cavity and release pathogen associated molecular patterns that interact with pathogen recognition receptors on local macrophages and neutrophils to initiate strong inflammatory reactions (*Iwasaki and Medzhitov, 2004*).

Macrophage activation is critically modulated by extracellular ATP and associated purinergic signalling via autocrine/paracrine mechanisms (*Idzko et al., 2014*). During sepsis, purinergic signalling regulates core inflammatory cell functions such as migration and inflammatory mediators production (*Idzko et al., 2014*; *Ledderose et al., 2018*; *Ledderose et al., 2016*). ATP depletion or blockade of purinergic receptors were shown to increase survival and improve outcome of sepsis in rodent models (*Cauwels et al., 2014*; *Csóka et al., 2015*). The initiating event triggering purinergic signalling is

the release of extracellular nucleotides and includes active vesicular and channel-mediated release in addition to the passive release from necrotic cells (*Dosch et al., 2018*). Active ATP release from inflammatory cells can occur both via vesicular exocytosis or via connexin or pannexin hemichannels, mainly Connexin-43 (CX43) and pannexin-1 (*Dosch et al., 2018*; *Junger, 2011*). In sepsis, it is still unknown which mechanisms mediate ATP release and if targeting these mechanisms would impact outcomes. In addition, it remains to be described which inflammatory cells are involved in ATP release and to what extend ATP determines autocrine/paracrine inflammatory cell regulation during sepsis.

By screening release mechanisms of ATP using small molecule blockers, we have identified CX43 to be critical in peritoneal sepsis. CX43 is a gap junction protein forming either gap junctions or hemichannels and has been shown to mediate ATP release from inflammatory cells including macrophages (*Csóka et al., 2015*; *Kang et al., 2008*). CX43 full knockout is perinatally lethal in mice due to cardiac malformation (*Eckardt et al., 2004*; *Liao et al., 2001*). Thus, in order to specifically explore the function of CX43 on inflammatory cells, we developed a conditional MAC-CX43 KO mouse (*Lyz2*cre/cre*Gja1*flox/flox), in which CX43 is specifically deleted on macrophages and neutrophils. We identified that peritoneal macrophages express high CX43 levels in mice and humans and that CX43 expression on macrophages critically controls ATP release, local and systemic inflammation, and ultimately survival in a model of abdominal peritonitis via the purinergic receptor P2Y1. Our findings suggest that CX43 represents a new potential target for sepsis therapy.

## Results

### Small peritoneal macrophages release extracellular ATP via Connexin-43 during peritonitis

Inflammatory cell populations present in the peritoneal cavity during peritonitis were characterised using CLP as a murine model of peritoneal sepsis and include infiltrating small peritoneal macrophages (SPM) and resident large peritoneal macrophages (LPM) (*Ghosn et al., 2010*; *Wang and Kubes, 2016*). Ten hours after CLP, SPM populated the peritoneal cavity while the fraction of LPM disappeared (*Figure 1A–B*, *Figure 1—figure supplement 1A–B*). Additionally, two large populations of neutrophils invaded the peritoneal cavity upon sepsis while dendritic cells disappeared. Given the observation that SPM specifically populate the peritoneal cavity after CLP, we tested the ability of macrophages to secrete ATP. Isolated C57BL/6 wild-type (WT) peritoneal macrophages actively released ATP in response to TLR-4 agonist LPS (1 µg/ml) as well as the TLR-2 agonist Pam3CSK4 in a dose-dependent manner (*Figure 1C–D*). ATP released in the extracellular space was promptly degraded by ecto-nucleotidases, as lower levels of extracellular ATP were observed in absence of treatment with ARL67156 trisodium salt, an ecto-ATPase inhibitor (*Figure 1—figure supplement 1C*).

To screen cellular mechanisms underlying LPS-induced ATP release, specific blockers of (hemi) channels and vesicular exocytosis were administered (*Figure 1E*). LPS-induced ATP release was blocked by carbenoxolone (combined pannexin/connexin blockade), 18-alpha-GA (global connexin blockade) and Gap27 (blockade of Connexin-43 (CX43)) but not by probenecid (pannexin channel blockade) and N-Ethylmaleimide (blockade of vesicular exocytosis) revealing a specific role of CX43 (*Figure 1E*). The relevance of CX43 is substantiated by dose-dependent unspecific (18-alpha-glycyrrhetinic acid) and specific (Gap27) CX43 blockade (*Figure 1F–G*, *Figure 1—figure supplement 1D*).

To understand the importance of CX43 in ATP release from macrophages and its impact on macrophage activation, we developed conditional MAC-CX43 KO (*Lyz2*cre/cre, *Gja1*flox/flox) mice in which CX43 is specifically deleted in activated macrophages (*Figure 1—figure supplement 2*). As expected from the previous results using a specific CX43 blocker, ATP was released from WT peritoneal macrophages upon LPS stimulation but not from MAC-CX43 KO peritoneal macrophages, while intracellular ATP levels remained constant (*Figure 1H–I*, *Figure 1—figure supplement 1E*).

### Connexin-43 expression is induced on macrophages in a MyD88/TRIF dependent manner

In humans, fluids obtained from peritoneal lavage during abdominal operations were analysed by flow cytometry. CX43high macrophages were observed in the peritoneal fluid of patients with

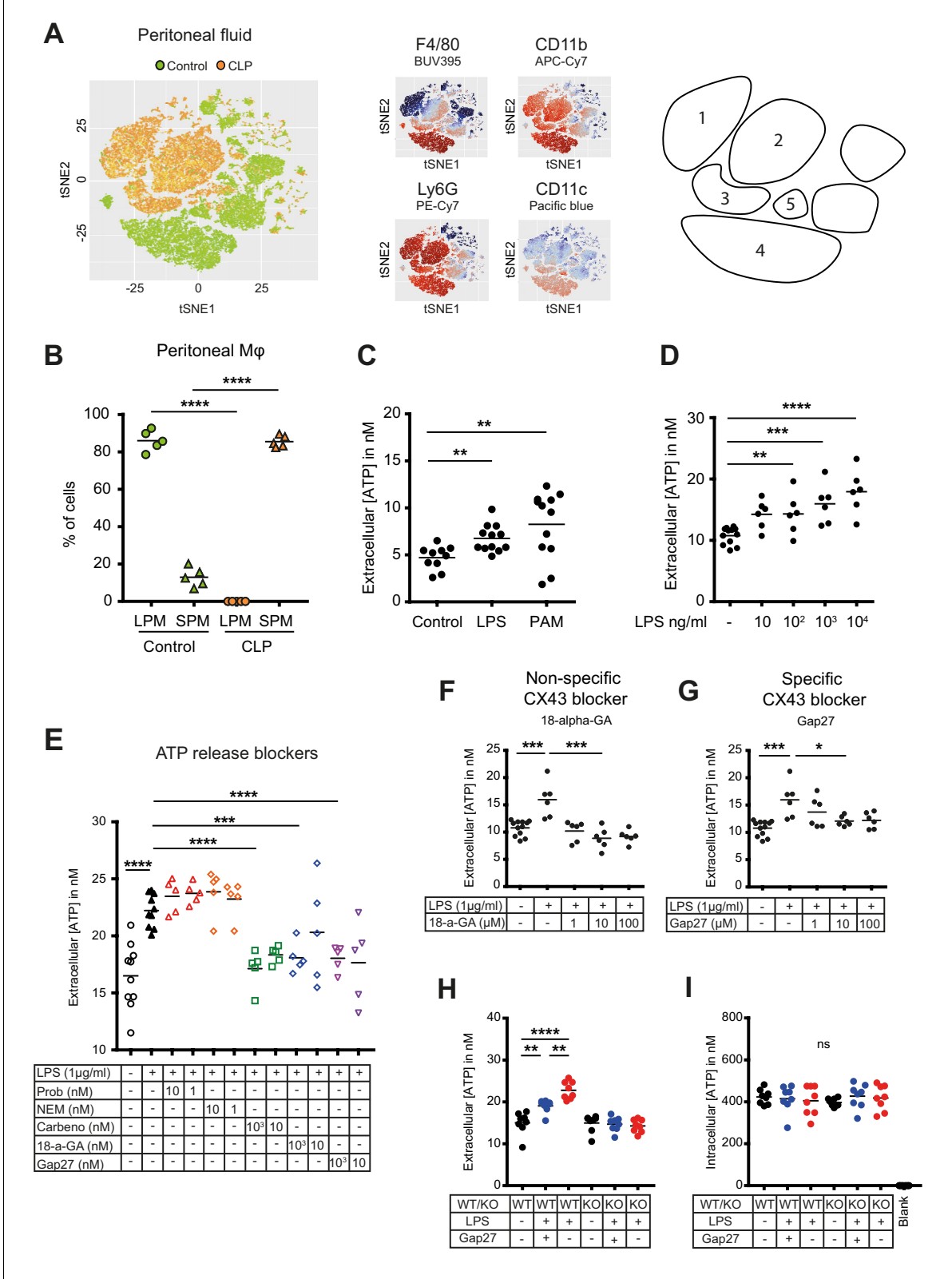

**Figure 1.** Small peritoneal macrophages release ATP during sepsis via Connexin-43 hemichannels. (**A**) Intraperitoneal cell fractions characterised by flow cytometry in controls (green) and 10 hr after caecal ligation and puncture (CLP, orange) in a tSNE plot of viable (AmCyan[low]) CD45[high] cells. Populations were defined based on F4/80 (BUV395), CD11b (APC/Cy7), Ly6G (PE-Cy7) and CD11c (Pacific blue). In response to CLP, elevated levels of neutrophils (1)/(2) (F4/80[low], CD11b[high], Ly6G[high]) and small (infiltrating) peritoneal macrophages (3) (F4/80[int], CD11b[int]) were observed but not large

*Figure 1 continued on next page*

*Figure 1 continued*

(resident) peritoneal macrophages (4) (F4/80$^{high}$, CD11b$^{high}$) or dendritic cells (5) (F4/80$^{low}$, CD11b$^{low}$, Ly6G$^{low}$, CD11c$^{high}$) (N = 5 animals per group). (**B**) Relative frequency of large (resident) peritoneal macrophages (LPM) and small (infiltrating) peritoneal macrophages (SPM) in the peritoneal cavity of controls (green) and 10 hr after CLP (orange) (N = 5 animals per group, one-way ANOVA). (**C**) Extracellular ATP levels in the supernatant of peritoneal macrophages after 30 min stimulation with LPS (TLR-4 agonist) or Pam3CSK4 (PAM, TLR-2 agonist). Cells were isolated from C57 Bl/6 WT mice (each dot is representative of an independent biological replicate, representative of more than five independent experiments, unpaired t-test). (**D**) Dose-dependent ATP release by peritoneal macrophages in response to LPS as quantified by a luciferin-luciferase assay (N = 12 (control) and 6 (LPS))*. (**E**) LPS-induced ATP release by macrophages can be blocked by carbenoxolone ((Carbeno) combined pannexin/connexin blockade), 18-alpha-GA (18-a-GA) global connexin blockade), Gap27 (specific blockade of Connexin 43 (CX43)) but not by probenecid ((Prob) pannexin channel blocker) and N-Ethylmaleimide ((NEM) blockade of vesicular exocytosis) (N = 10 (negative and positive controls) and 5 (blockers))*. (**F–G**) Non-specific (18-a-GA) and specific (Gap27) blocking of connexin hemichannels decreases LPS-induced ATP release from WT peritoneal macrophages in a dose-dependent manner (N = 12 (negative control) and 6 (treated groups), negative and positive controls were the same for both experiments)*. (**H**) LPS-induced ATP release from WT compared to MAC-CX43 KO peritoneal macrophages with and without specific CX43 blocking (N = 8)*. (**I**) LPS-induced ATP release has no impact on intracellular ATP levels (N = 8)*. *Data represent independent biological replicates, compared by unpaired t-test and are representative of three or more independent experiments.

DOI: https://doi.org/10.7554/eLife.42670.002

The following figure supplements are available for figure 1:

**Figure supplement 1.** Flow cytometry gating strategy for peritoneal inflammatory cells and characterization of ATP release from peritoneal macrophages.

DOI: https://doi.org/10.7554/eLife.42670.003

**Figure supplement 2.** Generation of conditional MAC-CX43 KO.

DOI: https://doi.org/10.7554/eLife.42670.004

peritonitis but were not detected in patients without signs of intra-abdominal infection (*Figure 2A–C*, *Supplementary file 1A*).

In mice, analysis of cellular fractions present in the peritoneal cavity during peritonitis revealed that CX43-dependent ATP release is macrophage-specific, since both SPM and LPM are CX43-positive, while neutrophils are CX43-negative (*Figure 2D*). CX43 protein expression in WT murine peritoneal macrophages is induced by LPS (*Figure 2E–F*, *Figure 2—figure supplement 1A–B*). Mechanistically, MyD88 and TRIF are required for the expression of CX43 after LPS stimulation (*Figure 2G–H*), while it is independent of caspase-1 and caspase-11 (*Figure 2I*). Thus, CX43 in SPM is responsible for LPS-induced ATP release in peritoneal sepsis in a MyD88/TRIF dependent manner.

## Connexin-43 expressing macrophages are recruited systemically during peritonitis

Sepsis is a systemic inflammatory reaction that leads to extensive organ damage remote from the primary infection site, typically in the liver and the lungs. To understand whether CX43 positive macrophages would be present locally at the site of infection only or in remote organs as well, we analysed hepatic and pulmonary tissues in our CLP model. In the liver of controls, CX43 was expressed mainly on parenchymal cells including cholangiocytes, while after CLP it was highly upregulated on infiltrating cells in a time-dependent manner (*Figure 3A–B*, *Figure 3—figure supplement 1A*). CX43 protein expression in the liver was dependent on MyD88/TRIF activation (*Figure 3C*). Following CLP, infiltrating macrophages and neutrophils increased in the liver while resident macrophages (Kupffer cells) remained constant (*Figure 3D–E*). CX43 expression in the liver was restricted to infiltrating macrophages, while neutrophils and resident macrophages were CX43 low (*Figure 3F*). Hepatic lympho-cellular populations, such as T cells, B cells, and NK cells did not express CX43 (*Figure 3—figure supplement 1B–D*).

In the lungs, CX43 is constitutively expressed on muscle cells and resident macrophages populations (*Figure 3G*). In the acute respiratory distress syndrome that is associated with severe peritonitis, alveolar infiltration of CX43 inflammatory cells was observed (*Figure 3G–H*). These results reveal that CX43 expression is increased in the liver and lungs as remote target organs during sepsis.

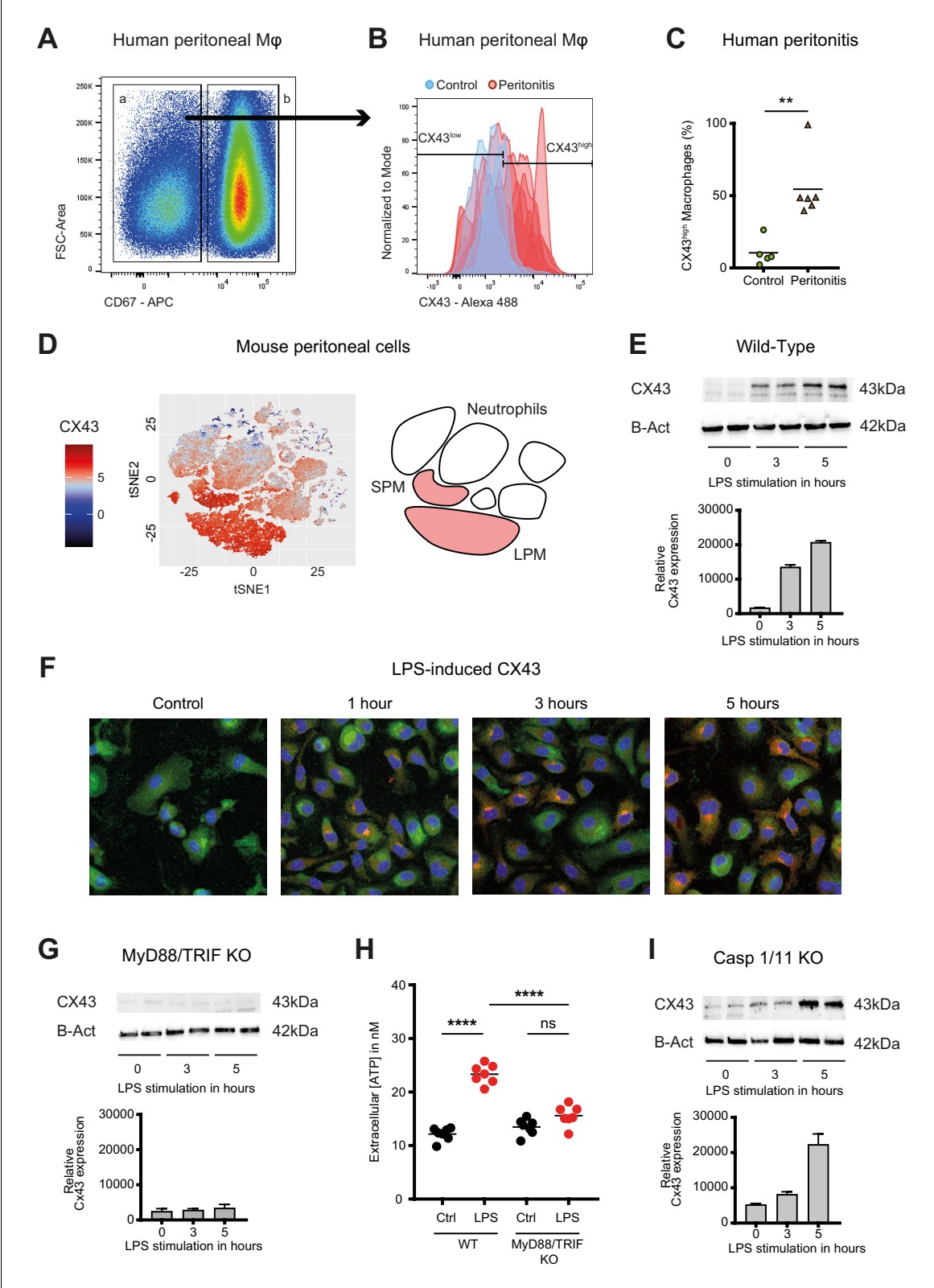

**Figure 2.** Connexin-43 expression is induced on macrophages in a MyD88/TRIF dependent manner. (**A**) Flow cytometry gating strategy for human inflammatory cells isolated from fluids obtained from peritoneal lavage during abdominal operations. Macrophages were defined as viable (AmCyan[low]), CD45[high], CD3[low], CD19[low], CD56[low], CD11c[low], CD16[high] and CD67[low] cells. (**B**) CX43 expression level on macrophages isolated from patients with peritonitis (red) compared to control patients (blue). (**C**) Percentage of CX43[high] macrophages in the peritoneal fluid collected from control

*Figure 2 continued on next page*

Figure 2 continued

patients without peritoneal inflammation (green) compared to patients with peritonitis (orange) (N = 5 patients in control group and N = 6 patients in peritonitis group, Mann-Whitney test). (D) Specific CX43 expression (Alexa Fluor 488) in macrophages (small and large peritoneal macrophages, SPM and LPM) among peritoneal viable (AmCyan$^{low}$) CD45$^{high}$ cells (N = 5 animals per group, tSNE analysis as described in statistical methods). (E–F) Increased expression of CX43 in peritoneal macrophages following stimulation with LPS (1 µg/ml) for the indicated time-points (Western blot: N = 2 independent biological replicates per time-point, quantification of immunoblots using ImageJ, unpaired t-test. Immunofluorescence: PE-Texas red = CX43; FITC = F4/80; blue = DAPI). (G) Abrogated CX43 protein expression in peritoneal macrophages from MyD88/TRIF double KO mice upon stimulation with LPS 1 µg/ml (N = 2 mice per time-point, quantification of immunoblots using ImageJ, unpaired t-test). (H) LPS-induced ATP release from murine WT compared to MyD88/TRIF KO peritoneal macrophages (N = 7 independent biological replicates, one-way ANOVA). (I) CX43 protein expression in peritoneal macrophages from caspase 1/11 double KO mice upon stimulation with LPS 1 µg/ml (N = 2 mice per time-point, quantification of immunoblots using ImageJ, unpaired t-test). Full uncut western blot membranes are available in Source data 1.

DOI: https://doi.org/10.7554/eLife.42670.005

The following figure supplement is available for figure 2:

**Figure supplement 1.** Connexin-43 expression on peritoneal macrophages.

DOI: https://doi.org/10.7554/eLife.42670.006

## Connexin-43-mediated local and systemic ATP release contributes to macrophage over-activation via P2Y1

Levels of extracellular ATP were increased in the peritoneal cavity as well as in the systemic circulation 10 hr after CLP (*Figure 4A–B*). Following CLP, we observed lower systemic ATP levels in MAC-CX43 KO mice compared to the WT mice, suggesting a role for CX43 in systemic ATP release, supporting the results in *Figure 1* showing that local ATP release from macrophages is CX43-dependent (*Figure 4B*). In a next step, we assessed autocrine responses of CX43-dependent ATP release in peritoneal macrophages. Pharmacological blocking (Gap27) or genetic deletion of CX43 decreased pro-inflammatory cytokines levels (TNF-alpha and IL-6) after 3 and 6 hr of LPS stimulation in peritoneal macrophages in vitro (*Figure 4C–D*). Gap27 decreased TNF-alpha levels after 6 hr LPS-stimulation and IL-6 levels after 3 hr LPS-stimulation. Next, we aimed to understand whether CX43 blocking or deletion, and consequent decrease in ATP release, regulate macrophage differentiation. Functional markers of M1 differentiation, such as *Inos* and *Il12rb*, were decreased in peritoneal macrophages under CX43 blocking or deletion (*Figure 4—figure supplement 1A–B*), while markers of M2 differentiation, including *Arg1*, *Tgfb* and *Il-10*, were comparable among groups (*Figure 4—figure supplement 1C–E*). Phagocytosis of IgG coated latex beads was not impacted by CX43 blocking or deletion (*Figure 4—figure supplement 1F*). Therefore, autocrine CX43-dependent ATP signalling alters cytokine secretion and drives M1 macrophage differentiation but not phagocytosis.

To determine if cytokine secretion in response to abrogated ATP release after CX43 deletion can be reverted, ATPgammaS, a non-hydrolyzable form of ATP, was administered in vitro. Thereby the inhibited secretion of IL-6 by peritoneal macrophages was restored to control levels (*Figure 4E*, *Figure 4—figure supplement 1G*). Administration of apyrase, a soluble ecto-ATPase consuming extracellular ATP, decreased pro-inflammatory cytokines levels. Thus, the downstream effects of LPS-induced CX43-dependent ATP release is mediated via purinergic receptors. To identify specific P2 receptors responsible for this effect, a screen using different P2 receptors blockers was performed by measuring the effects of LPS-dependent TNF-alpha and IL-6 release from peritoneal macrophages (Source data 1). The observed reversal of TNF-alpha and IL-6 secretion following unspecific P2 receptor blockade (suramin) and specific P2Y1 blockade (MRS 2279) indicate a crucial role of P2Y1 in macrophage activation following LPS stimulation (*Figure 4F–G*, *Figure 4—figure supplement 2A–B*). Gene expression of purinergic ATP receptors (P2X and P2Y receptors) were not differently regulated between MAC-CX43 KO and WT (*Figure 4—figure supplement 2C*). Extracellular ATP is hydrolysed by the ecto-nucleotidase CD39 to ADP/AMP and by ecto-5'nucleotidase CD73 to adenosine. CD39 mRNA levels were reduced in response to CX43 pharmacological blockade or genetic deletion in MAC-CX43 KO peritoneal macrophages compared to WT controls, while no difference between WT and MAC-CX43 KO was observed for ecto-5'nucleotidase CD73 (*Figure 4—figure supplement 3A–B*). However, differences in CD39 mRNA expression between WT and MAC-CX43 KO peritoneal macrophages had no impact on kinetics of extracellular ATP degradation by these cells (*Figure 4—figure supplement 3C–D*). Taken together, CX43 blockade or deletion reduce ATP secretion and its autocrine downstream effects on macrophages via P2Y1.

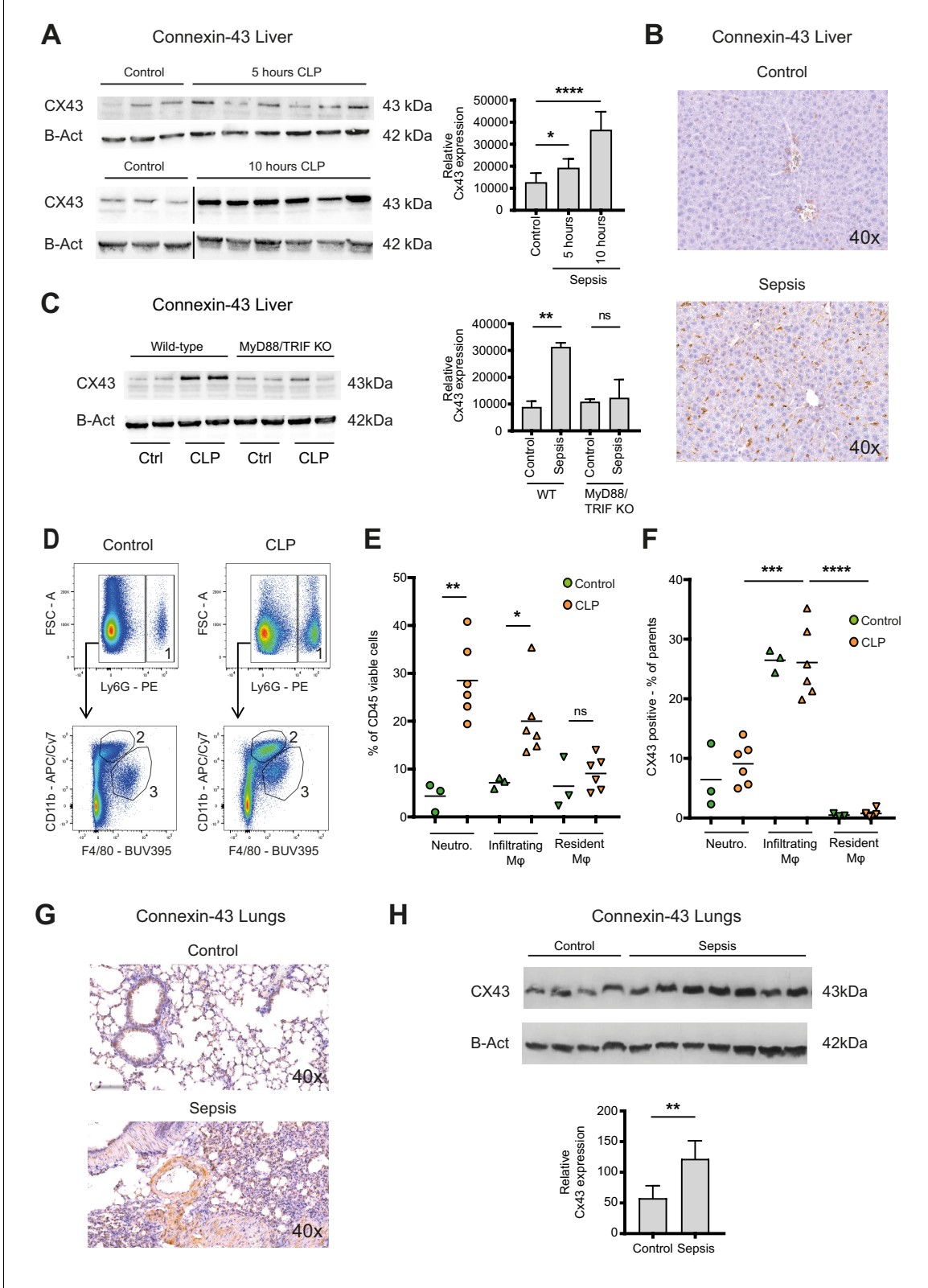

**Figure 3.** Connexin-43 expressing macrophages are recruited systemically during sepsis. (**A**) Elevated CX43 protein expression in the liver 5 and 10 hr after CLP compared to controls (N = 3 animals in each control group from the gels at 5 hr and 10 hr, all used for the calculation of the histogram) and N = 6 animals for each time point, unpaired t-test). (**B**) Elevated expression of CX43 on non-parenchymal cells in the liver 10 hr after CLP compared to controls. (**C**) Abrogated CX43 protein expression in the liver of MyD88/TRIF double KO mice 10 hr after CLP (N = 2 mice per group, unpaired t-test).
*Figure 3 continued on next page*

*Figure 3 continued*

(D–E) Following CLP (orange), neutrophils (1) and infiltrating macrophages (2) increased in the liver, while resident macrophages (3) remained constant compared to controls (green). (F) Infiltrating macrophages highly expressed CX43 levels while neutrophils (1) and resident macrophages (3) were CX43 low (N = 3 animals in control group and N = 6 animals in CLP group, unpaired t-test). (G, H) Elevated expression of CX43 on non-parenchymal cells in the lungs in response to CLP compared to control mice (N = 4 animals in control group and N = 7 animals in CLP group, unpaired t-test). Full uncut western blot membranes are available in Source data 1.

DOI: https://doi.org/10.7554/eLife.42670.007

The following figure supplement is available for figure 3:

**Figure supplement 1.** Connexin-43 expression in the liver.
DOI: https://doi.org/10.7554/eLife.42670.008

## Improved survival and decreased local and systemic cytokine secretion in response to Connexin-43 blocking or deletion during abdominal sepsis

To test the relevance of CX43 for sepsis outcome, we compared our MAC-CX43 KO mice ($Lyz2^{cre/cre}$, $Gja1^{flox/flox}$) to mice heterozygous for the lyzozyme two cre sequence ($Lyz2^{cre/wt}$, $Gja1^{flox/flox}$) with or without pharmacological inhibition with Gap27 as well as sham operated controls. Murine sepsis score (*Supplementary file 1C*), weight loss, survival time, bacterial load, and markers of inflammation were monitored. The extent of sepsis, as assessed by the murine sepsis score, was significantly reduced in MAC-CX43 KO compared to WT mice without a difference in weight loss (*Figure 5A–B*). CLP is a lethal model with a survival time expected to be reached between 18 and 24 hr following surgery. CX43 blocking or specific genetic deletion in CLP operated mice was associated with significantly prolonged survival time compared to control WT C57BL/6 mice (*Figure 5C*). Bacterial load was not different in both peritoneal fluid and blood excluding altered phagocytosis by macrophages to be relevant (*Figure 5—figure supplement 1A–B*).

In order to assess if cellular fractions change in MAC-CX43 KO mice compared to WT mice during sepsis, peritoneal cells were characterised by mass and flow cytometry. At baseline, two resident macrophages clusters (LPM) were identified (*Figure 5—figure supplement 2A–C*). The number of CX43$^{high}$ resident macrophages was higher in the peritoneal cavity of WT mice compared to MAC-CX43 KO mice at baseline, which might be due to low Lyz2 promoter activation during their maturation (*Figure 5—figure supplement 2C*). LPM disappeared 10 hr after CLP and neutrophils and different clusters of infiltrating SPM were increased in the peritoneal cavity (*Figure 5—figure supplement 2D*). However, no difference in cell numbers was observed between WT and MAC-CX43 KO mice, indicating that even though macrophages were differently activated, their numbers were not altered by CX43 deletion or blockade. Relative fractions of CD4 and CD8 T cells decreased in septic animals compared to control without difference between WT and MAC-CX43 KO (*Figure 5—figure supplement 3A–B*).

Inflammatory cytokines were assessed systemically (serum), locally (peritoneal fluid), and in key organs typically injured during sepsis (lungs, liver, kidney, intestine). In the serum, we observed decreased levels of pro-inflammatory cytokines (TNF-alpha and IL-33 but not IL-6 and IL-1 beta) in MAC-CX43 KO compared to WT mice 10 hr after CLP (*Figure 5D–F*, *Figure 5—figure supplement 3C*). In addition, IL-10 levels were lower, indicating a decrease of both pro- and anti-inflammatory cytokines in MAC-CX43 KO mice (*Figure 5G*). In the peritoneal fluid and in the peripheral effector organs, we observed decreased IL-6 levels (with the exception of the intestine) of mice with CX43 deletion or blockade compared to non-treated WT mice (*Figure 5H–K*, *Figure 5—figure supplement 3D*). A relevant role of neutrophils can be excluded, first, because of limited expression of CX43 in response to sepsis (*Figures 2 and 3*) and because no alterations of myeloperoxidase activity in the liver of mice 10 hr after CLP was observed in response to CX43 blockade (*Figure 5—figure supplement 3E–F*).

Taken together these data suggest that CX43 regulates the secretion of inflammatory cytokines in an autocrine manner during sepsis by the release of ATP and activation of P2Y1.

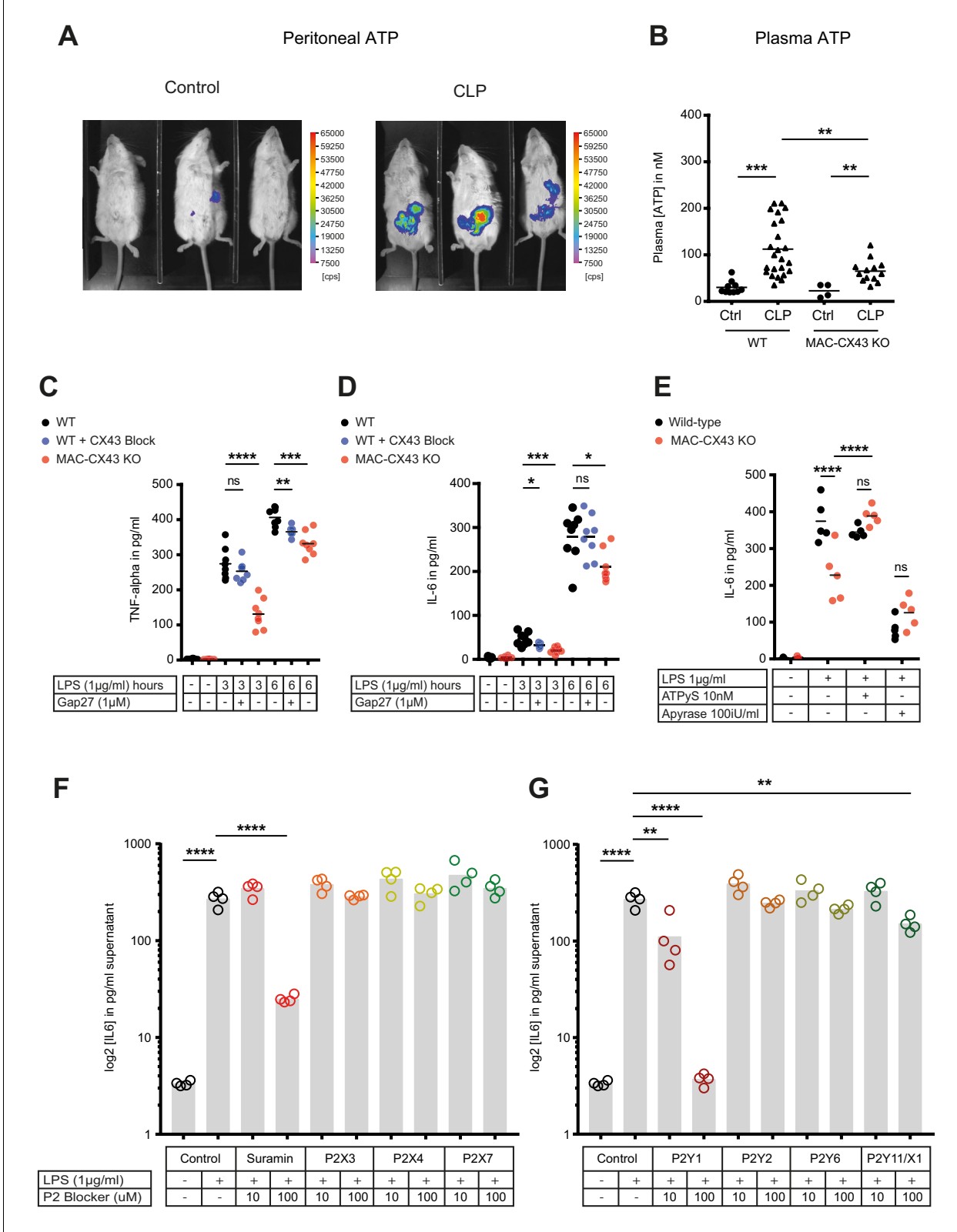

**Figure 4.** Connexin-43-mediated local and systemic ATP release contributes to macrophage over-activation via P2Y1. (A) Extracellular ATP levels as assessed by luminescence of i.p. injected 5 × 10⁶ HEK293-pmeLUC D-luciferin activated cells (in counts per second (cps)). Luminescence was significantly higher 10 hr after CLP compared to sham-operated controls. (B) Direct measurement of extracellular ATP in the plasma collected from vena cava inferior is elevated in response to CLP and higher in WT mice compared to MAC-CX43 KO mice (each dot is representative for a single animal,
*Figure 4 continued on next page*

*Figure 4 continued*

unpaired t-test). (C–D) Inhibition (Gap27 (1 µM)) or genetic deletion of CX43 decreased TNF-alpha (C) und IL-6 (D) secretion from peritoneal macrophages in response to stimulation with LPS (1 µg/ml) for 3 and 6 hr (removal of 7 outliers after ROUT test in *Figure 4D*)*. (E) IL-6 release was restored by activation of purinergic receptors by exogenous administration of ATPγS (10 nM) and abrogated in response to apyrase (100 IU/ml) (each dot is representative of an independent biological replicate, two-way ANOVA, representative of three experiments). (F–G) IL-6 release from peritoneal macrophages in response to LPS (1 µg/ml) and by blocking major P2X (G) and P2Y (H) purinergic receptors (N = 4)*. *Data represent independent biological replicates, compared by unpaired t-test and are representative of three or more independent experiments.

DOI: https://doi.org/10.7554/eLife.42670.009

The following figure supplements are available for figure 4:

**Figure supplement 1.** Characterization of peritoneal macrophage activation.

DOI: https://doi.org/10.7554/eLife.42670.010

**Figure supplement 2.** Regulation of P2-type receptors on peritoneal macrophages.

DOI: https://doi.org/10.7554/eLife.42670.011

**Figure supplement 3.** No consequences of Connexin-43 deletion on function of CD73 and CD39 in peritoneal macrophages.

DOI: https://doi.org/10.7554/eLife.42670.012

## Discussion

Following hollow organ perforation, intestinal bacteria penetrate the peritoneal cavity and generate inflammation eventually leading to sepsis (*Figure 6*). Our data have shown that ATP release from peritoneal macrophages is an active process that occurs in response to TLR-2 and TLR-4 agonists in a CX43-dependent manner. Increased expression of CX43 on macrophages following TLR-4 pathway activation has been previously shown (*Qin et al., 2016*). Our study has extended this observation by describing the functional role of CX43 for autocrine signalling in macrophages. This is mainly relevant in infiltrating SPM, which are the main CX43-positive population during peritonitis, replacing resident LPM.

Once released in the extracellular space via CX43, ATP typically signals through P2 receptors in an autocrine manner. Among the P2 receptors that have been shown to play a critical role for macrophage activation (*Csóka et al., 2015*; *Greve et al., 2017*; *Lecut et al., 2012*; *Maître et al., 2015*), we identified the P2Y1 to be critical in a model of abdominal peritonitis. This effect mainly depends on alterations in cytokine secretion while phagocytic capacity was not changed, adding evidence to previous conflicting results on putative purine mediated phagocytosis (*Anand et al., 2008*; *Glass et al., 2013*).

We have shown that both pharmacologic blockade and genetic deletion of CX43 in activated macrophages led to increased survival compared to controls. This was associated with lower levels of key inflammatory cytokines in the acute phase of sepsis including IL-6, TNF-alpha in response to CX43 blockade or deletion. Decreased levels of IL-10 and IL-33 indicate that blocking CX43 might improve the long-term outcomes, as these cytokines have been shown to be involved in immunosuppression following sepsis (*Nascimento et al., 2017*). This supports previous data showing that PAMPs and DAMPs interacting with PRRs induce ATP release and finally alterations in cytokine secretion (*Cauwels et al., 2014*; *Sakaki et al., 2013*). This effect was inflammasome independent for TNF-alpha and IL-6. Furthermore, the finding that IL-1-beta secretion is not altered via the CX43/P2Y1 axis supports inflammasome independent signalling (*Gicquel et al., 2015*; *Sakaki et al., 2013*). P2Y1 purinergic receptors are $G_{\alpha q}$-protein coupled receptors signalling via phospholipase Cβ and increased cytosolic calcium levels (*Barańska et al., 2017*). The function and downstream signalling of P2Y1 has been well described in platelets or neurons but not in macrophages (*Hechler and Gachet, 2015*). It is likely that downstream signalling of P2Y1 mediated TNF-alpha and IL-6 secretion in macrophages to be dependent on the same mechanisms. CX43/P2Y1 dependent modulation of IL-6 secretion may be partially dependent on TNF-alpha since IL-6-promoter is also activated by TNF-alpha among other stimuli explaining the sequence of appearance during peritonitis (*Hunter and Jones, 2015*; *Jones and Jenkins, 2018*).

In this study, plasma and peritoneal ATP levels might be underestimated in vivo, since ATP is rapidly degraded by different membrane-bound ecto-nucleotidases (NTPDases 1, 2 and 8) and soluble ecto-nucleotidases (*Fausther et al., 2012*; *Feldbrügge et al., 2018*; *Yegutkin et al., 2003*). This is supported by the finding that by blocking ecto-nucleotidases, we observed higher ATP levels in the extracellular space of peritoneal macrophages in vitro (*Figure 1—figure supplement 1C*). Altered

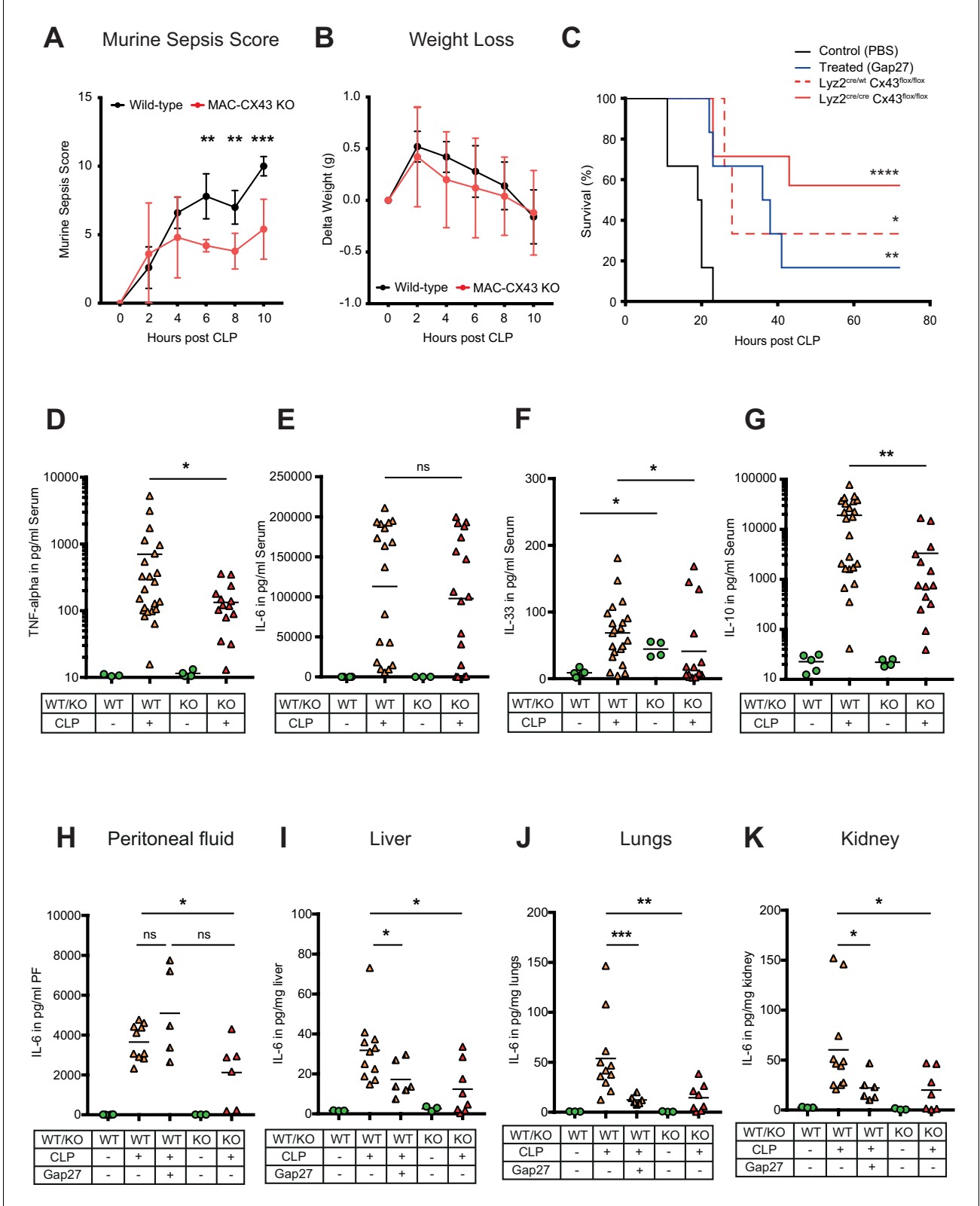

**Figure 5.** Improved survival and decreased local and systemic cytokine secretion in response to Connexin-43 blocking or deletion during abdominal sepsis. (A–B) Clinical outcome using Murine Sepsis Score and weight loss following CLP (N = 5 animals per group, two-way ANOVA). (C) Survival of mice following caecal ligation and puncture (CLP). Mice treated with Gap27 (blue, N = 6 animals), a specific CX43 blocker, homozygous (*Lyz2*$^{cre/cre}$, *Gja1*$^{flox/flox}$, red, N = 7 animals) and heterozygous cre (*Lyz2*$^{cre/wt}$, *Gja1*$^{flox/flox}$, red dashed, N = 3 animals) were compared to non-treated WT controls

*Figure 5 continued on next page*

*Figure 5 continued*

(black, N = 6 animals) (Log-rank (Mantel-Cox) test). (D–G) Systemic levels of TNF-alpha, IL-6, IL-33 and IL-10 in the serum of WT and MAC-CX43 KO mice operated with CLP (each dot is representative of a single animal, Mann-Whitney test). (H–K) IL-6 in peritoneal fluid (PF) serum, lungs, kidney and liver (each dot is representative of a single animal, Mann-Whitney test). Levels are expressed in pg/mg tissue respectively pg/ml of serum or peritoneal fluid. Data are representative of three or more independent experiments.

DOI: https://doi.org/10.7554/eLife.42670.013

The following figure supplements are available for figure 5:

**Figure supplement 1.** No consequences of Connexin-43 deletion on local and systemic bacterial load.
DOI: https://doi.org/10.7554/eLife.42670.014
**Figure supplement 2.** Impact of CX43 on inflammatory cell recruitment to the peritoneal cavity.
DOI: https://doi.org/10.7554/eLife.42670.015
**Figure supplement 3.** Consequences of peritonitis on distribution of T cells subsets, IL1beta, IL-6 and MPO.
DOI: https://doi.org/10.7554/eLife.42670.016

purinergic signalling by MAC-CX43 KO peritoneal macrophages resulting in decreased cytokine secretion was reversed by the administration of ATPgammaS or apyrase. Therefore, CX43 may impact on ATP dependent autocrine signalling via activation or desensitisation of purinergic receptors that has been shown for P2Y1 (*Gomes et al., 2009*; *Rodríguez-Rodríguez et al., 2009*). The expression of CD39 was disturbed in response to CX43 deletion as seen in other models of

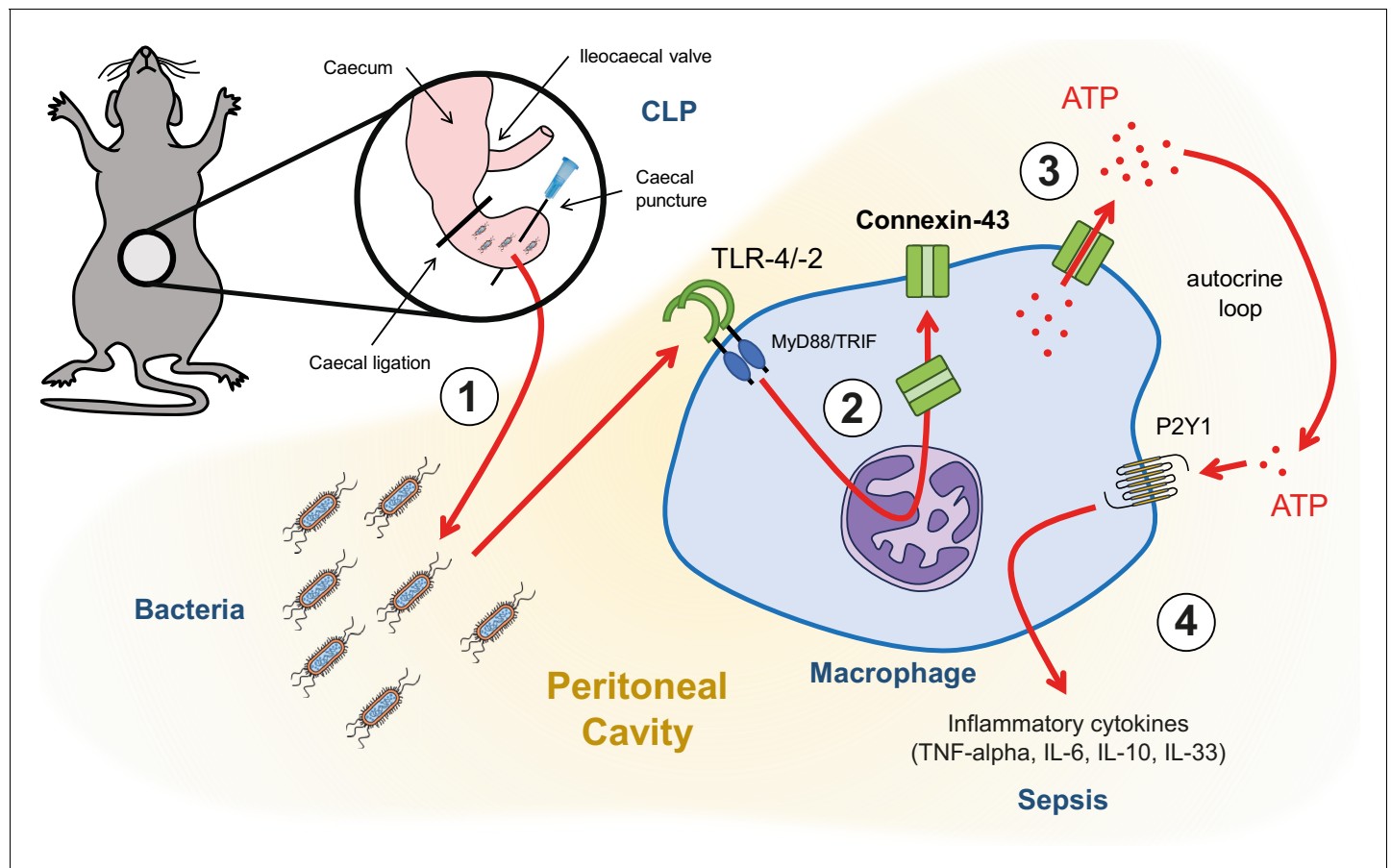

**Figure 6.** Graphical summary. (1) Following caecal ligation and puncture (CLP), bacteria invade the peritoneal cavity. (2) Bacteria-derived products, for example LPS, interact with TLR-4 to increase CX43 synthesis in a MyD88/TRIF dependent manner and (3) with TLR-4/–2 to induce CX43-dependent ATP release. (4) Extracellular ATP interacts with P2Y1 receptors to increase inflammatory cytokines secretion.
DOI: https://doi.org/10.7554/eLife.42670.017

inflammation (*Deaglio and Robson, 2011*). However, altered expression had no impact on the kinetics of ATP hydrolysis in cultured peritoneal macrophages.

The effects of CX43 observed in this study are specific for macrophages given the exclusive expression of CX43 on macrophages conversely to neutrophils or lymphocytes. Therefore, we present a novel pharmacological target to address macrophage function specifically. Clinical applicability is substantiated by human data where we identified CX43 positive macrophages in the peritoneal cavity of patients with peritonitis but not in control patients, allowing us to hypothesise that CX43 plays a comparable role in human macrophages and in mouse macrophages.

As CX43 proteins form pores that are permeable for other small molecules, typically lower than 1 kDa, the effect on the secretion of other nucleotides, including UDP (*Qin et al., 2016*), as well as other metabolites could influence sepsis outcome as well.

For in vitro experiments, thioglycollate was used to induce recruitment of macrophages to the peritoneal cavity. Thioglycollate-induced exudate macrophages differ from resident macrophages in metabolic activity and phagocytic activity (*Pavlou et al., 2017*; *Shaw and Griffin, 1982*). However, given the robust differences between the groups we expect that the findings are applicable to SPM that are the recruited and highly active macrophage population in peritonitis.

In conclusion, we identified that CX43 is upregulated specifically on macrophages locally and systemically during sepsis and that the elevated levels of CX43 mediate cytokine secretion but not phagocytosis of macrophages via extracellular ATP and the purinergic receptor P2Y1. Therefore, CX43 is critical for the pathophysiology of sepsis and may be a potential new target for improving the outcome of this critical clinical condition.

# Materials and methods

**Key resources table**

| Reagent type (species) or resource | Designation | Source or reference | Identifiers | Additional information |
|---|---|---|---|---|
| Antibody | Anti-Connexin-43 (rabbit, polyclonal) | Cell Signalling | 3512, RRID:AB_2294590 | WB (1:850) IHC/IF (1:100) |
| Antibody | Anti-β-actin HRP-conjugated (mouse, monoclonal) | Sigma-Aldrich | A2228, RRID:AB_476697 | (1:5000) |
| Antibody | Anti-rabbit HRP-conjugated (goat, polyclonal) | Dako | P0448, RRID:AB_2617138 | (1:5000) |
| Antibody | Anti-rabbit biotinylated (goat, polyclonal) | Dako | E0432, RRID:AB_2313609 | (1:200) |
| Antibody | Anti-rabbit IgG FITC-conjugated (goat, polyclonal) | Sigma-Aldrich | F0382, RRID:AB_259384 | (1:200) |
| Cell line (*Homo sapiens*) | HEK293-pmeLUC | DOI: 10.1371/journal.pone.0002599 | | |
| Chemical compound | ARL 67156 | Sigma-Aldrich | 67156 | |
| Chemical compound | TRIzol | Roche | 11667165001 | |
| Chemical compound | Omniscript RT Kit 200 | Quiagen | 205113 | |
| Chemical compound | Lipopolysaccharides E.coli O111:B4 | Sigma-Aldrich | L4130 | |
| Chemical compound | Pam3CSK4 | Invivogen | tlrlpms | |
| Chemical compound | 18-alpha-glycyrrhetinic acid | Sigma-Aldrich | G8503 | |

*Continued on next page*

*Continued*

| Reagent type (species) or resource | Designation | Source or reference | Identifiers | Additional information |
|---|---|---|---|---|
| Chemical compound | Gap27 | Tocris-Bioscience | 1476 | |
| Chemical compound | Probenecid | Sigma-Aldrich | P8761 | |
| Chemical compound | Carbenoxolone | Sigma-Aldrich | C4790 | |
| Commercial assay | ATP Reagent SL | BioThema | 144–041 | |
| Commercial assay | Resazurin assay | Sigma-Aldrich | 199303 | |
| Commercial kit | V-PLEX Mouse Cytokine 19-Plex kit | Meso Scale Discovery | K15255D-1 | |
| Commercial kit (ELISA) | TNF-alpha | Invitrogen | 88-7324-22 | |
| Commercial kit (ELISA) | IL-6 | Invitrogen | 88-7064-22 | |
| Commercial kit (ELISA) | IL-10 | Invitrogen | 88-7105-22 | |
| Commercial kit (ELISA) | IL-1 beta | Invitrogen | 88-7013-86 | |
| Commercial assay | Phagocytic assay | Cayman Chemical | 500290 | |
| Other | Brewer thioglycollate | Sigma-Aldrich | B2551 | |
| Other | DMEM/F12 GlutaMAX | ThermoFisher Scientific, Gibco | 10565018 | |
| Other | CTAD tubes | BD Vacutainer | 367599 | |
| Other | Peroxidase labelled streptavidin | Seracare | 71-00-38 | (1:200) |
| Other | DAPI stain | Sigma-Aldrich | D9542 | (1:1000) |
| Other | Ficoll | GE Healthcare | 17-5442-02 | |
| Other | Myeloperoxidase assay | doi:10.1097/TP.0000000000000101 | | |
| Strain, strain background (*Mus musculus*) | Wild-type mice | Harlan, Netherlands | C57BL/6J | |
| Strain, strain background (*Mus musculus*) | MyD88/TRIF double-knockout mice | Mucosal Immunology, University of Bern | B6.129-MyD88tm1Aki x B6-Ticam1lps2 | |
| Strain, strain background (*Mus musculus*) | Caspase-1/11 double-knockout mice | Mucosal Immunology, University of Bern | B6N.129S2-Casp1tm1Flv | |
| Strain, strain background (*Mus musculus*) | MAC-CX43 knockout mice | Visceral Surgery, University of Bern | B6.129P2-Lyz2tm1(cre)Ifo/J x B6.129S7-Gja1tm1Dlg/J | Cell-specific conditional MAC-CX43 KO mouse model developed in our laboratory |
| Strain, strain background (*Mus musculus*) | Lyz2-cre mice | Jackson Laboratory | B6.129P2-Lyz2tm1(cre)Ifo/J | |
| Strain, strain background (*Mus musculus*) | Gja1 floxed mice | Jackson Laboratory | B6.129S7-Gja1tm1Dlg/J | |

## Human samples

Peritoneal lavage fluids were collected from patients operated at the Department of Visceral Surgery and Medicine, Inselspital Bern, Switzerland (*Supplementary file 1A*). Peritoneal lavage fluid was

filtered through a 100 μm filter and a Ficoll (GE Healthcare, #17-5442-02) gradient was performed by pipetting 10 ml Ficoll under 40 ml peritoneal lavage fluid in a 50 ml Falcon tube. Tubes were centrifuged (800 g, 4°C, 20 min, no brake). Cells were stained for flow cytometry as described below with anti-human antibodies (*Supplementary file 1E*).

## Experimental animals

Animals were housed in specific pathogen free (SPF) conditions. Animals used in our experiments were aged between 8 to 12 weeks and were females. WT mice were on C57Bl/6(J) background and were purchased from Harlan, Netherlands. MyD88/TRIF$^{-/-}$ mice and caspase 1/11$^{-/-}$ mice were provided by Andrew MacPherson (Mucosal Immunology, University of Bern) and were bred germ-free at the Clean Mouse Facility, University of Bern, Switzerland. These mice were colonised with SPF flora for experimental purposes.

Due to the fact that CX43 is expressed on various cell types including endothelial cells and myocytes in the heart in addition to inflammatory cells, CX43 full KO die perinatally due to cardiac malformation (*Eckardt et al., 2004*; *Liao et al., 2001*). In order to develop a cell-specific conditional MAC-CX43 KO mouse model, B6.129P2-Lyz2tm1(cre)Ifo/J mice and B6.129S7-Gja1tm1Dlg/J mice were purchased from Jackson Laboratory (Bar Harbor, ME, USA) and were crossed together. MAC-CX43 KO mice used in our experiments were carrying a floxed Gja1 gene on both alleles and cre enzyme either on both alleles (*Lyz2$^{cre/cre}$*, *Gja1$^{flox/flox}$*) or on one allele for heterozygous controls (*Lyz2$^{cre/wt}$*, *Gja1$^{flox/flox}$*). Genotyping of MAC-CX43 KO mice was done according to Jackson protocol and using following primers: Gja1 (Forward primer: 5' CTTTGACTCTGATTACAGAGCTTAA 3' and reverse primer: 5' GTCTCACTGTTACTTAACAGCTTGA 3') and Lyz2 (Wild-type forward primer: 5' TTACAGTCGGCCAGGCTGAC 3', mutant (cre) forward primer: 5' CCCAGAAATGCCAGATTACG 3', common reverse primer: 5' CTTGGGCTGCCAGAATTTCTC 3').

## Caecal ligation and puncture

We used caecal ligation and puncture (CLP) to induce peritonitis leading to high-grade sepsis (*Rittirsch et al., 2009*). In brief, mice were anaesthetised with isoflurane and a longitudinal 1 cm midline laparotomy was performed to expose the caecum. The caecum was tightly ligated using a 6.0 silk suture (6–0 PROLENE, Ethicon) with 2/3 of the caecum included below the ligature and was perforated twice using an 22 gauge needle. The abdomen was closed and mice were subcutaneously injected with 1 ml of 0.9% saline. Sham-operated mice received the same surgical operation without caecum ligation and puncture. No antibiotics were administered to mice that underwent the operation. The mice were sacrificed at 5 and 10 hr for experimental purposes. A separated set of animals, including WT and MAC-CX43 KO, were used in survival studies and were sacrificed latest 72 hr after CLP. All the animals operated with CLP were monitored using a Murine Sepsis Score that was adapted from a previous publication (*Shrum et al., 2014*) (*Supplementary file 1C*). Animals were treated with analgesics (buprenorphin) or sacrificed according to their score.

## Collection of blood, peritoneal fluid and organs

Prior to harvesting, mice were anaesthetised using 2 μl/kg mouse body weight of a mixed triple combination of Fentanyl (0.05 mg/ml), Midazolam (5 mg/ml) and Medetomidin (1 mg/ml). Animals were placed on a surgical tray, abdominal skin was removed and laparotomy was performed. Blood was collected from inferior vena cava using a 22 gauge catheter (BD Insyte, #381223) and a 2 ml syringe into 1.5 ml Eppendorf tubes. Blood was incubated at room temperature for 60 min and centrifuged at 1,000 g for 10 min. Supernatant was transferred in a new tube and further centrifuged at 10'000 g for 10 min. Serum was finally collected in a new tube and snap frozen in liquid nitrogen. In order to collect peritoneal fluid, 5 ml ice cold PBS was injected i.p. using a 10 ml syringe and a 23 gauge needle. The abdomen was massaged for 15 s and at least 1 ml peritoneal fluid was collected in Eppendorf tubes. Tubes were centrifuged at 10,000 g for 10 min to pellet cells present in the peritoneal fluid and supernatant was pipetted into a new tube. Both pellets and supernatants were snap frozen in liquid nitrogen for further experiments. Various organs, including liver, lungs, kidney and small intestine were collected and either further processed for flow cytometry analyses or snap frozen in liquid nitrogen for future experiments.

## Isolation and culture of mouse peritoneal macrophages

Mice were injected i.p. with 2 ml 3% Brewer thioglycollate (Sigma-Aldrich, #B2551) 3 days prior macrophage isolation. In order to collect macrophages, the peritoneal cavity was washed with ice cold harvest medium (Dulbecco's phosphate-buffered saline without calcium and magnesium supplemented with 3% fetal calf serum, Sigma-Aldrich, #D8537). Collected fluids from at least five different mice were pooled and filtered through 100 μm filter and cells were centrifuged at 800 g and 4°C for 5 min. Erythrocytes were lysed using erythrocyte lysis buffer (Qiagen, #160011922). Cells were counted using a counting chamber and were plated after resuspension in culture medium (Dulbecco's Modified Eagle Medium/F12 GlutaMAX (ThermoFisher Scientific, Gibco, #10565018) supplemented with 50 IU/ml penicillin G and 50 μg/ml streptomycin and 10% fetal calf serum). HEK293-pmeLUC cells were provided by Francesco Di Virgilio (Department of Morphology, Surgery and Experimental Medicine, University of Ferrara). HEK293-pmeLUC cells were cultured in DMEM/F12 supplemented with 10% fetal calf serum, 50 IU/ml penicillin G, 50 μg/ml streptomycin and 0.2 mg/ml G418 sulphate (Geneticin, Calbiochem). Cells were cultured in 75 cm$^2$ tissue culture flasks (VWR, #734–2313) and kept in the incubator at 37°C and 5% $CO_2$. The viability of cultured cells was checked under the microscope before every cell passage. After detachment of cells using a 0.05% trypsin/EDTA solution (ThermoFisher Scientific, #15400054), cells were counted using a counting chamber (Brand, #717805). $40 \times 10^3$ cells were plated on 96-wells transparent flat bottomed cell culture plates (Sarstedt, Tissue culture plate 96-wells, sterile, #83.3924.300) and $1 \times 10^6$ cells were plated on 6-wells transparent flat bottomed cell culture plates (Sarstedt, Tissue culture plate 6-wells, # 83.3920.005) for experimental purposes.

## Quantification of extracellular and intracellular ATP

Extracellular ATP was measured using a luciferin-luciferase assay as described in manufacturer protocol (Biothema, #144–041) and bioluminescence was quantified using a Tecan Infinite 200 plate reader (Tecan). In vitro, the supernatant was collected from 96-wells cell culture plate. The ecto-ATPase inhibitor ARL 67156 (Sigma-Aldrich, #67156) was used to reduce extracellular ATP degradation. In vivo, extracellular ATP was quantified in the peritoneal fluid and in the plasma using the same kit and 1:20 dilution in PBS. Plasma was collected in Eppendorf tubes with CTAD (citrate-theophylline-adenosine-dipyridamole) (BD Vacutainer, #367599) to avoid platelet activation and ATP release via degranulation. Intracellular ATP was quantified in peritoneal macrophages using a luminescent cell viability assay following manufacturer's protocol (CellTiter-Glo 2.0 Assay, Promega, # G9241).

## In vivo ATP imaging using plasma-membrane targeted luciferase

Measurements of extracellular ATP using a chimera plasma-membrane targeted luciferase on HEK293 cells were performed as previously described and by adapting the protocol for our own purposes (*Pellegatti et al., 2008*; *Pellegatti et al., 2005*). Cells were cultured as described above. $5 \times 10^6$ cells were injected i.p. right after CLP or sham and before suturing the abdominal wall. 6 hr after CLP, 3 mg D-luciferin diluted in 100 μl PBS were injected i.p. Bioluminescence was measured using a gamma camera (NightOWL LB 983, Berthold Technologies) 8 min after D-luciferin injection and is proportional to the amount of ATP present in the peritoneal cavity.

## Quantification of Colony forming units (CFUs) in blood and peritoneal lavage

As previously described (*Csóka et al., 2015*), blood and peritoneal lavage fluid were diluted serially in sterile physiologic saline. Fifty microliters of each dilution was aseptically plated and cultured on LB agar or trypticase blood agar plates (BD Biosciences) at 37°C. After 12–16 hr of incubation, the number of bacterial colonies was counted. The number of cultures is expressed as CFUs per millilitre of blood or peritoneal lavage fluid.

## Gene expression analysis

Total RNA was isolated from tissue by TRIzol reagent and following manufacturer's protocol (Roche, #11667165001). In the liver, one cubic millimetre of the median lobe was used for RNA isolation. RNA concentration and quality were analysed by spectrophotometer NanoDrop ND-1000 (Thermo

Scientific). 500 ng of total RNA was used for cDNA synthesis. cDNA was synthesised by using Omniscript RT Kit 200 (Quiagen, #205113). qPCR analysis was performed by using TaqMan gene expression assays (Applied Biosystem) according to manufacturer's protocol. mRNA was analysed by reverse transcriptase-quantitative PCR (ABI 7900, SDS 2.3 software). Primers and probes sequences were purchased from Applied Biosystems: *Arg1* (Mm00475988_m1), *Cd39/Entpd1* (Mm00515447_m1), *Cd73* (), *Il-10* (Mm00439614_m1), *Inos* (Mm00440485_m1), *Il-12rb2* (Mm00434200_m1), *P2r × 1* (forward: 5' ACTGGGAGTGTGACCTGGAC 3', reverse: 5' TCCCAAACACCTTGAAGAGG 3'), *P2r × 2* (forward: 5' CAAAGCCTATGGGATTCG 3', reverse: 5' CCTATGAGGAGTTCTGTT 3'), *P2r × 3* (forward: 5' ATTAAGATCGGCTGGGTGTG 3', reverse: 5' TCCCGTATACCAGCACATCA 3'), *P2r × 4* (forward: 5' TACGTCATTGGGTGGGTGTT 3', reverse: 5' CTTGATCTGGATACCCATGA 3'), *P2r × 5* (forward: 5' GGGGTTCGTGTTGTCTCTGT 3', reverse: 5' CACTCTGCAGGGAAGTGTCA 3'), *P2r × 6* (forward: 5' GTGGTAGTCTACGTGATAGG 3', reverse: 5' GCCTCTCTATCCACATACAG 3'), *P2r × 7var1-3* (forward: 5' CTTGCCAACTATGAACGG 3', reverse: 5' CTTGGCCTTTGCCAACTT 3'), *P2r × 7var4* (forward: 5' TCACTGGAGGAACTGGAAGT 3', reverse: 5' TTGCATGGATTGGGGAGCTT 3'), *P2ry1* (forward: 5' TTATGTCAGCGTGCTGGTGT 3', reverse: 5' CGTGTCTCCATTCTGCTTGA 3'), *P2ry2* (forward: 5' GAGGACTTCAAGTACGTGCT 3', reverse: 5' ACGGAGCTGTAAGCCACAAA 3'), *P2ry4* (forward: 5' AACAACTGCTTCCTCCCT 3', reverse: 5' AAGTCCTAGAGGTAGGTG 3'), *P2ry6* (forward: 5' CCTGATGTATGCCTGTTCAC 3', reverse: 5' CACAGCCAAGTAGGCTGTCT 3'), *P2ry11* (forward: 5' TGTGGCCCATACTGGTGGTTGAG 3', reverse: 5' GAAGAAGGGGTGCACGATGCCCA 3'), *P2ry12* (forward: 5' ATATGCCTGGTGTCAACACC 3', reverse: 5' GGAATCCGTGCAAAGTGGAA 3'), *P2ry13* (forward: 5' TGCAGGGCTTCAACAAGTCT 3', reverse: 5' CCTTTCCCCATCTCACACAT 3'), *P2ry14* (forward: 5' GGAACACCCTGATCACAAAG 3', reverse: 5' TGACCTTCCGTCTGACTCTT 3'). Relative changes in mRNA were calculated with the ΔΔCt method. Ct values of target genes were calculated relative to a reference control gene (*Tbp* (Mm01277042_m1) and Beta-actin (*Actb*, Mm00607939_s1)) using the following formula ΔCtTG = CtTG CtRG. Experimental groups are normalised to control group. ΔΔCt=ΔCtexp - ΔCtcon. Fold change = 2-ΔΔCt Log2FC.

## Immunoblot

Proteins were extracted using RIPA Lysis buffer, which volume was adapted for every tissue. Proteins extracted were quantified using Bio-Rad Protein Assay System (Bio-Rad Laboratory, Melville, NY). Protein lysates were boiled in Laemmli buffer, run on BIO-RAD Mini-Protean TGXTM (10–20% Ready Gel Tris-HCl Gel System, 12-well comb #456–1095 and 10-well comb #456–1094) for 90 min at 120 Volt, and transferred on BIO-RAD Trans-Blot Turbo Mini or Midi PVDF membranes (Mini PVDF Transfer Packs #170–4156 and Midi PVDF Transfer Packs #170–4159) by semi dry Transfer (Trans-Blot Turbo Transfer System BIO-RAD). Specific proteins were detected using the following primary antibodies: Connexin-43 (1:850) (Cell Signalling, #3512, RRID:AB_2294590), β-actin HRP-conjugated (1:5000) (Sigma-Aldrich, #2228, RRID:AB_476697). Antibodies were diluted in 5% milk and incubation was done overnight at 4°C. Following secondary antibodies were used: HRP-conjugated anti-rabbit (1:5000) (Dako, #P0448, RRID:AB_2617138). Membranes were then washed and protein expression was analysed by chemiluminescence (Western Lightning Plus-ECL Perkin Elmer), using Fusion-FX (Vilber).

## Immunohistochemistry and immunofluorescence

CX43 expression in the liver and in the lungs was assessed by immunostaining. In brief, paraffin-embedded tissue sections were dewaxed followed by incubation in antigen retrieval citrate buffer (Sigma-Aldrich, #C9999). Erythrocyte quenching was performed using 3% $H_2O_2$ (Dr. Grogg, #K45052100407) diluted in phosphate buffered saline supplemented with 0.05% Tween 20 (PBST). After protein blocking (PBS, 5% BSA), the slides were incubated overnight at 4°C with the primary rabbit anti-CX43 antibody (1:100, Cell Signalling, #3512), followed by incubation with secondary biotinylated anti-rabbit (Dako, #E0432) for 1 hr at room temperature and by incubation with peroxidase labelled streptavidin (Seracare, #71-00-38) for 30 min at room temperature. Liver and lung sections were stained with hematoxylin (MERK, #HX43078349) for 1 min followed by Eosin staining for 6 min. Later, sections were mounted with Eukritt (Kindler, GmbH).

Peritoneal macrophages were isolated as described above and plated on a chamber slide for immunofluorescence. After incubation at 37°C for 12 hr, cells were washed using PBS and fixed using 4% formaldehyde solution for 15 min. Specimens were covered with blocking buffer (1X PBS/5% normal serum/0.3% Triton X-100) for 60 min followed by overnight incubation with primary rabbit anti-CX43 antibody (1:100, Cell Signalling, #3512, RRID:AB_2294590). Further, specimens were incubated 120 min with secondary goat anti-rabbit IgG FITC-conjugated antibody (Sigma, #F0382, RRID: AB_259384) and 5 min in Dapi (Sigma, #D9542). Finally, specimens were mounted using Vectashield Antifade Mounting Medium (Vector Laboratories, #H-1000) and covered with a coverslip.

## Cytokine measurement

Inflammatory cytokines (TNF-alpha, IL-6 and IL-10) were quantified in serum, peritoneal fluid and organs collected in mice following CLP and in control mice, and supernatants from in vitro LPS-stimulated peritoneal macrophages. Uncoated ELISA kits were purchased from Invitrogen (Thermo Fisher Scientific) and ELISA were performed following manufacturer's protocol: TNF-alpha (Invitrogen, #88-7324-22), IL-6 (Invitrogen, #88-7064-22), IL-10 (Invitrogen, #88-7105-22). Plates were red using Tecan.

Serum cytokines were measured using electrochemiluminescence assays (Meso Scale Discovery (MSD), Rockville, Maryland, USA). Serum TNF-alpha, IL-6, IL-10 and IL-33 were measured using a multiplex assay using the V-PLEX Mouse Cytokine 19-Plex kit (MSD). Samples were diluted 1:3 in proprietary buffer (MSD) and measured in duplicate according to manufacture's protocol. Plates were run on MSD plate reader model 1250.

## Neutrophil detection by myeloperoxidase assay

Myeloperoxidase assay was carried out as previously described (*Fahrner et al., 2014*; *Mota et al., 2007*). Briefly, liver samples were thawed, 50 mg of tissue was weighed out and homogenised in 1 ml of 20 mM phosphate buffer using a TissueLyser (Qiagen) 2 min at 20 Hz, centrifuged at 10,000 g for 10 min, and the pellet was resuspended in 50 mM phosphate buffer (pH 6.0) containing 0.5% hexadecyltrimethyl ammonium bromide. The samples were then freeze-thawed four times using liquid nitrogen followed by a 40 s sonication, centrifuged at 10,000 g for 10 min, and the supernatant was used for the MPO assay. The assay mixture contained 10 µl of the sample, 25 µl 3,3'5,5-tetramethylbenzidine (final concentration 1.6 mM), 25 µl $H_2O_2$ in 80 mM phosphate buffer (pH 5.4, final concentration $H_2O_2$ 0.3 mM), and 40 µl of 50 mM phosphate buffer (pH 6.0) containing 0.5% hexadecyltrimethyl ammonium bromide. The mixture was incubated at 37°C for 2 min, stopped with the addition of 1 M HCl, and the absorbance was measured at 450 nm.

## Flow cytometry

To perform flow cytometry, leukocytes were isolated from the peritoneal cavity and from the liver. Isolation of peritoneal leukocytes was performed as described above by washing the peritoneal cavity with ice cold PBS. Leukocytes isolation from the liver was performed as previously described (*Kudira et al., 2016*). The liver was cut in small pieces and placed in 10 ml RPMI containing 3% FCS 1 mg/ml collagenase IV and 0.1 mg/ml DNase I (Roche) with shaking at 37°C for 30 min. Cell suspension was passed through a 100 µm cell strainer and washed twice with FACS buffer (PBS, 3% FCS, 2 mM EDTA). Cells were centrifuged (800 g, 7 min, 4°C) and resuspended in 40% Percoll solution and layered on top of 80% Percoll solution. Gradient centrifugation was carried out (900 g, 20 min, 4°C, no brake). Leukocytes were collected from the interphase, washed with FACS buffer and centrifuged (800 g, 7 min, 4°C). Cells were finally resuspended in FACS buffer and further stained for flow cytometry.

Aliquots of $10^6$ cells/100 µL of staining buffer per well were incubated each with 1 µg of purified anti-CD16/CD32 for 20 min at 4°C in the dark, in order to block non-specific binding of antibodies to the FcγIII and FcγII receptors. Cell suspension was incubated with a fixable viability dye (AmCyan, eBioscience, #65-0866-14) diluted in DPBS during 20 min at 4°C in the dark to exclude dead cells. Subsequently, these cells were separately stained with the surface markers for 20 min at 4°C in the dark with 1 µg of primary antibodies. For cytokines and transcription factors, cells were stained with antibodies to surface antigens, fixed and permeabilised according to the manufacturer's instructions (Foxp3/Transcription Factor Staining Buffer Set; eBioscience, 00-5523-00). Fluorescently labelled

anti-mouse used are summarised below (*Supplementary file 1D*). Cells were subsequently washed twice with, and re-suspended in FACS buffer. Finally, cell data were acquired on a LSR II SORP H271 (BD Biosciences). Flow cytometric analysis was done using FlowJo (Treestar). In all experiments, FSC-H versus FSC-A was used to gate on singlets with dead cells excluded using the fluorescence-coupled fixable viability dye. Murine inflammatory peritoneal cells were gated as follow: (1) SPMs: viable (AmCyan$^{low}$), CD45$^{high}$, CD3 and CD19$^{low}$, Ly6G$^{low}$, F4/80$^{int}$ and CD11b$^{int}$ (2) LPMs: viable (AmCyan$^{low}$), CD45$^{high}$, CD3 and CD19$^{low}$, Ly6G$^{low}$, F4/80$^{high}$ and CD11b$^{high}$ (3) Neutrophils: viable (AmCyan$^{low}$), CD45$^{high}$, CD3 and CD19$^{low}$, Ly6G$^{high}$, F4/80$^{low}$ and CD11b$^{low}$ (4) Dendritic cells: viable (AmCyan$^{low}$), CD45$^{high}$, CD3 and CD19$^{low}$, Ly6G$^{high}$, F4/80$^{low}$, CD11b$^{low}$ and CD11c$^{high}$. Liver macrophages were characterised as published previously (*You et al., 2013*). Inflammatory cells in the liver were gated as follow: AmCyan$^{low}$, CD45$^{high}$, CD3/CD19$^{low}$.

Human peritoneal macrophages were gated as follow: AmCyan$^{low}$, CD45$^{high}$, CD3$^{low}$, CD19$^{low}$, CD56$^{low}$, CD11c$^{low}$, CD16$^{high}$ and CD67$^{low}$.

## Time-of-flight mass cytometry (CyTOF)

For time-of-flight mass cytometry (CyTOF) experiments, cells were isolated from the peritoneal cavity as previously describe (*Ray and Dittel, 2010*), $3 \times 10^6$ cells were stained with lanthanide-conjugated antibodies as previously described (*Becher et al., 2014*). Antibodies were purchased already labelled or labelled using the MaxPar antibody labelling kit (Fluidigm, *Supplementary file 1F*). Raw signals were normalised to median bead intensity using four elemental calibration beads (Fluidigm) as previously described (*Finck et al., 2013*). The Live intact single events were gated for their DNA content (MaxPar Intercalator, Ir 193) and viability (Cell-ID Cisplatin, 198 Pt). CD45$^{neg}$ cells were excluded. Analysis was done using t-SNE for dimensionality reduction on arc sinus transformed data (*Hinton, 2008*). On t-SNE plots the clusters were manually gated based on their marker expression.

## Phagocytic assay

The phagocytic capacity of plated peritoneal macrophages was assessed as described in the manufacturer protocol (Cayman Chemical, #500290). Peritoneal macrophages were isolated from WT and MAC-CX43 KO mice as described above. 50'000 macrophages were plated per well on a chamber slide (Nunc Lab Tek II, #154534) and incubated for 12 hr at 37°C and 5% CO$_2$. Macrophages were primed or not with LPS 1 µg/ml for 4 hr and exposed or not to FITC latex beads coated with rabbit IgG diluted 1:200 for 2 hr (four groups in total). Negative controls were incubated on a separate slide chamber at 4°C.

## Compounds

Toll-like receptors 4 and 2 agonists were used to stimulate peritoneal macrophages in vitro: lipopolysaccharides (LPS) from E.coli O111:B4 (Sigma-Aldrich, # L4130) and Pam3CSK4 (Invivogen, # tlrlpms). In order to block ATP release via hemichannels, 18-alpha-glycyrrhetinic acid (Sigma-Aldrich, # G8503) was used as a nonspecific inhibitor of connexin channels and Gap27 (Tocris Bioscience, # 1476) was used as a specific CX43 inhibitor. 18-alpha-glycyrrhetinic acid was dissolved in DMSO to obtain a solution 20 mg/ml as described in supplier protocol. The final DMSO concentration in the supernatant was 0.25%.

Probenecid (Sigma-Aldrich, # P8761) was used as a specific pannexin-1 inhibitor and carbenoxolone (Sigma-Aldrich, # C4790) as a non-specific inhibitor of connexin and pannexin channels.

The following purinergic agonists and antagonists were used: ATPgammaS (Jena Bioscience, # NU-406–50), apyrase (Sigma-Aldrich, # A6132), suramin (non-selective P2 antagonist, Tocris, # 1472); NF110 (P2 $\times$ 3 antagonist, Tocris, # 2548); 5BDBD (P2 $\times$ 4 antagonist, Tocris, # 3579); A804598 (P2 $\times$ 7 antagonist, Tocris, # 4473); MRS2500 and MRS2279 (P2Y1 antagonist, Tocris, # 2159 and # 2158); AR-C 118925XX (P2Y2 antagonist, Tocris, # 4890); MRS2578 (P2Y6 antagonist, Tocris, # 2146); NF157 (P2Y11/P2 $\times$ 1 antagonist, Tocris, # 2450). 5-BDBD, A804598, AR-C 118925XX and MRS2578 were dissolved in DMSO to obtain a solution 50 mM as described in supplier protocol. The final DMSO concentration in the supernatant was 0.2%.

## Statistical analysis

Survival statistics were determined by the Kaplan-Meier curve and log-rank test. t-test, Mann-Whitney test and ANOVA were used to assess the significance of the differences in the means of the different populations. When necessary for ANOVA, p-values were corrected for multiple comparisons using Turkey correction. Statistical tests were performed using Prism six software (GraphPad Software). Levels of significance were assessed with specifically indicated tests and p-values are presented as follows: ns p non-significant; *p<0.05; **p<0.01; ***p<0.001 and ****p<0.0001. Sample size calculation was performed prior to experiments on ClinCalc.com using statistical methods described in a previous publication (*Rosner, 2011*). t-SNE visualizations were produced with R-package Rtsne on arc sinus transformed data (*Van Der Maaten and Hinton, 2008*).

## Study approval

All human studies were approved by the Ethical Commission of the Canton Bern and written informed consent was obtained from all patients. Peritoneal fluid collection at the beginning of an operation was included in a larger clinical trial, whose protocol is published on ClinicalTrials.gov (NCT03554148, Study ID Number: Bandit).

Animal experiments were planned, carried out and reported in agreement with current 3R and ARRIVE guidelines (*Kilkenny et al., 2010*) and approved according to Swiss animal protection laws by the Veterinary Authorities of the Canton Bern, Switzerland (license no. BE 4/15).

# Acknowledgments

The authors would like to thank collaborators from the Laboratory of Visceral and Transplantation Surgery, Department for BioMedical Research, University of Bern for technical assistance. Mass cytometry was performed at the Cytometry Facility of the University of Zurich.

# Additional information

### Funding

| Funder | Grant reference number | Author |
|---|---|---|
| Schweizerischer Nationalfonds zur Förderung der Wissenschaftlichen Forschung | 323530_158117 | Michel Dosch |
| Novartis Stiftung für Medizinisch-Biologische Forschung | 14C160 | Michel Dosch |
| University of Bern | Interdisciplinary Grant | Guido Beldi |
| Schweizerischer Nationalfonds zur Förderung der Wissenschaftlichen Forschung | 166594 | Guido Beldi |
| Schweizerischer Nationalfonds zur Förderung der Wissenschaftlichen Forschung | 146986 | Guido Beldi |

The funders had no role in study design, data collection and interpretation, or the decision to submit the work for publication.

### Author contributions

Michel Dosch, Conceptualization, Resources, Data curation, Software, Formal analysis, Supervision, Funding acquisition, Validation, Investigation, Visualization, Methodology, Writing—original draft, Project administration, Writing—review and editing; Joël Zindel, Conceptualization, Data curation, Software, Formal analysis, Validation, Investigation, Methodology, Writing—review and editing; Fadi Jebbawi, Conceptualization, Formal analysis, Validation, Investigation, Methodology, Writing—review and editing; Nicolas Melin, Conceptualization, Validation, Investigation, Methodology, Writing—review and editing; Daniel Sanchez-Taltavull, Data curation, Software, Formal analysis, Methodology, Writing—review and editing; Deborah Stroka, Conceptualization, Resources, Supervision,

Funding acquisition, Validation, Project administration, Writing—review and editing; Daniel Candinas, Conceptualization, Resources, Supervision, Funding acquisition; Guido Beldi, Conceptualization, Resources, Data curation, Formal analysis, Supervision, Funding acquisition, Validation, Investigation, Visualization, Methodology, Writing—original draft, Project administration, Writing—review and editing

### Author ORCIDs
Michel Dosch (iD) http://orcid.org/0000-0002-4087-4293
Guido Beldi (iD) http://orcid.org/0000-0002-9914-3807

### Ethics
Human subjects: All human studies were approved by the Ethical Commission of the Canton Bern and written informed consent was obtained from all subjects. Peritoneal fluid collection at the beginning of an operation was included in a larger clinical trial, whose protocol is published on ClinicalTrials.gov (NCT03554148, Study ID Number: 2017-00573).

Animal experimentation: Animal experiments were planned, carried out and reported in agreement with current 3R and ARRIVE guidelines (Kilkenny et al., 2010) and approved according to Swiss animal protection laws by the Veterinary Authorities of the Canton Bern, Switzerland (license no. BE 4/15).

### Decision letter and Author response
Decision letter https://doi.org/10.7554/eLife.42670.022
Author response https://doi.org/10.7554/eLife.42670.023

# Additional files

### Supplementary files
• Supplementary file 1. Anti-mouse antibodies used for Time-of-Flight Mass Cytometry. (**A**) Diagnosis of human patients included in the study. (**B**) P2 receptors antagonists. (**C**) Visceral Surgery Murine Sepsis Score Sheet. (**D**) Anti-mouse antibodies used for flow cytometry (**E**) Anti-human antibodies used for flow cytometry (**F**) Anti-mouse antibodies used for Time-of-Flight Mass Cytometry
DOI: https://doi.org/10.7554/eLife.42670.018

• Source data 1. Full uncut western blot membranes.
DOI: https://doi.org/10.7554/eLife.42670.019

• Transparent reporting form
DOI: https://doi.org/10.7554/eLife.42670.020

### Data availability
All data generated or analysed during this study are included in the manuscript and supporting files.

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
