## [Decision Letter]

Thank you for submitting your article "Connexin-43-dependent ATP release mediates macrophage activation during peritonitis" for consideration by *eLife*. Your article has been reviewed by three peer reviewers, including Jos van der Meer as the Reviewing Editor and Reviewer #1, and the evaluation has been overseen by Tadatsugu Taniguchi as the Senior Editor. The following individual involved in review of your submission has agreed to reveal their identity: Gennady Yegutkin (Reviewer #2).

The reviewers have discussed the reviews with one another and the Reviewing Editor has drafted this decision to help you prepare a revised submission.

This manuscript describing the role of connexin 43 (CX43) in macrophage activation in peritonitis and sepsis clearly has merits:

1) It convincingly demonstrates the role of CX43 after TLR stimulation and its role in ATP release and cytokine production.

2) The use of mice (a conditional MAC-CX43 KO mouse, in which CX43 was specifically deleted on macrophages and neutrophils; cecal ligation and puncture) and the validation in humans with peritonitis is adequate.

3) It demonstrates the kinetics of the mononuclear phagocytes and the systemic activation in remote organs (liver and lung).

4) The interventions (both pharmacological and genetic) are well in place and lead to the pivotal role of the purinergic receptor P2Y1.

The findings provide evidence that CX43 is critical for the pathophysiology of sepsis and may represent a new potential new target for improving the outcome of this critical clinical condition.

Major comments:

1) The experiments in WT, as compared with MAC-CX43 KO mice, revealed massive ATP release via CX43 channels. This should be accompanied by partial depletion of cellular ATP stores or alternatively, CX43-positive macrophages are characterized by higher ATP turnover rate. The former possibility can be tested experimentally, by comparing total intracellular ATP levels in resting versus LPS-stimulated WT and MAC-CX43 KO macrophages, and also in exudate versus resident peritoneal macrophages.

2) ATP released in the extracellular space was shown to be promptly degraded by ecto-nucleotidases, as higher levels of extracellular ATP were observed in the presence of ecto-ATPase inhibitor ARL67156 (Figure 1—figure supplement 1C). Along with NTPDase1/CD39-positive macrophages, other ecto-nucleotidases contribute to the metabolism of ATP in the tissues. For instance, in the liver, NTPDase2 and NTPDase8 are reportedly expressed on portal fibroblasts and bile canaliculi (Fausther et al., 2012; Feldbrügge et al., 2018). Moreover, the presence of soluble ATP-inactivating enzymes freely circulating in the bloodstream (e.g. nucleotide pyrophosphatase/ phosphodiesterase-1 (NPP1): Yegutkin et al., 2003) may lead to the underestimation of actual ATP levels measured in the systemic circulation (and probably, in the peritoneal cavity as well). These alternative and auxiliary inactivating pathways should be mentioned in the text discussed accordingly.

3) Data on markedly down-regulated CD39 mRNA levels in Gap27-treated and CX43-KO LPS-stimulated macrophages (Figure 4F) are interesting but somehow surprising. What might be the mechanism underlying this decrease: does it mean that CD39 expression is regulated by extracellular concentrations of the available ATP substrate? Please, amend or clarify. These data should also be ascertained using direct bioluminescent enzymatic assay and measuring the rate of exogenous ATP decay in control versus treated cells.

4) In the second paragraph on ATP release via CX43 and MyD88/TRIF, the conclusion is that CX43 is responsible for ATP-release via MyD88/TRIF. However, although the authors show that CX43 expression is lower in MyD88/TRIF KO mice, they do not show that ATP release is downregulated in these cells, and therefore one cannot draw this conclusion based on the available data. Did the authors include an experiment in which ATP release of MyD88/TRIF KO cells was measured, thereby linking the two observations?

5) Figure 2F shows that CX43 protein expression in primary murine macrophages is induced by LPS. Perhaps, higher quality/resolution/magnification images would be helpful for better visualization of CX43 staining pattern and subcellular localization in the treated macrophages. Along with the merged image, the authors should provide single channel stainings for Texas Red-labelled CX43 and FITC-labelled macrophages, maybe by showing the representative images only for control cells and those stimulated with LPS for 5 hours.

6) It is unclear to the reviewers how all the ex-vivo/in-vitro experiments with mouse peritoneal macrophages were conducted; it is not clarified in the figure legends or Materials and methods. Is each dot representing 1 mouse? Were all the mouse cells pooled? Why the experiments not conducted in a paired manner using cells from 1 mouse in the different conditions? How many cells were seeded per well?

In the same vein, which cells are used is not always specified. What for example does% parents mean in Figure 3F?

Such details are important for potential repeating and understanding the figures.

7) Why were the peritoneal macrophages isolated using thioglycollate? This changes the phenotype of the macrophages tremendously. As it is a potent oxygen scavenger, results should be interpreted with caution if ROS production plays a role. This should be discussed at least in the Discussion.

8) Looking at Figure 1, in panel C-F, the control group (which should be the same experiment all the time) shows a mean ATP release of 5, 10, 16 and 10 nM respectively. Can the authors explain this? Again, the difference could not be understood from the methods, as they were lacking.

9) In Figure 1E and F and also Figure 4G and H: solvent controls are missing. Some of the used inhibitors are dissolved in water and do not need a solvent control, but many are not dissolvable in water (such as 18-alpha-glycyrrhetinic acid (chloroform or DMSO), carbenoxolone (in NaOH), P2X4, P2X7 and P2Y2-inhibitors (in DMSO)). How were the inhibitors dissolved and did the authors include solvent controls?

10) In Figure 3A, the authors state to have used the same controls for the two western blots. The control blots look very different to me however. Can the authors comment on this? Furthermore, the authors state that CX43 was expressed on cholangiocytes, but the data are missing. Also, they mention Figure 3F twice, stating it shows different things. Is there a figure panel missing? Finally, again in this paragraph, they draw a conclusion which cannot be made on the available data. CX43 is indeed expressed in the liver and lungs as remote target organs, as they show. However, there is no data showing that CX43 is not expressed in other organs. Therefore, the conclusion that CX43 is specifically expressed in these organs is misleading.

11) Where the authors studied the expression of pro-inflammatory cytokines, they state some results, which the figures do not confirm. For example, the authors state that Gap27 decreases pro-inflammatory cytokine levels TNF-alpha and IL-6, but only IL-6 is decreased, whereas TNF-alpha is not. The same holds true for ATPgammaS reverting the CX43 blockade: TNF-alpha is not downregulated upon the knockout or Gap27 inhibition, and ATPgammaS does not revert this phenotype, since it is simply absent. Finally, in the assessment of systemic cytokines, again the authors state that IL-6 is decreased, but the data do not support this conclusion.

12) How CX43 expression enhances TNF and IL-6 production does not become clear. Is this transcriptional or translational? In addition, isn't the IL-6 not just secondary to TNF? This should at least be discussed if not further supported by experimental data.

13) In line with the previous comment: there is quite some literature (especially Anna Rubartelli's work) showing the role of ATP in the release of IL-1beta out of the macrophage. Since IL-1beta is another deleterious cytokine acting in concert with TNF, but only partially secondary to TNF production, it is a missed chance that the investigators did not include IL-1beta measurements and include the dynamics of that cytokine.

14) What is missed in the Discussion is a more crisp description on the sequence of events, starting with the entry of bacteria in the sterile peritoneal cavity.

15) The description of the mononuclear phagocytes is confusing: infiltrating monocytes, monocyte-derived macrophages, monocytes, small macrophages, resident peritoneal macrophages, primary peritoneal macrophages.

Conceptually, when a bacterial challenge in the peritoneal cavity occurs is that resident macrophages do the first job. Directly followed by the rapid neutrophil response and the slower influx of monocytes (which become exudate macrophages). The term primary macrophages for thioglycollate-induced exudate macrophages is confusing.

---

## [Author Response]

Major comments:1) The experiments in WT, as compared with MAC-CX43 KO mice, revealed massive ATP release via CX43 channels. This should be accompanied by partial depletion of cellular ATP stores or alternatively, CX43-positive macrophages are characterized by higher ATP turnover rate. The former possibility can be tested experimentally, by comparing total intracellular ATP levels in resting versus LPS-stimulated WT and MAC-CX43 KO macrophages, and also in exudate versus resident peritoneal macrophages.

We thank the reviewers for raising this important question. Therefore, we quantified both extracellular and intracellular ATP with or without LPS stimulation in WT and MAC-CX43 KO peritoneal macrophages. Extracellular ATP levels upon LPS stimulation were elevated in the extracellular space in WT but not MAC-CX43 KO peritoneal macrophages (new Figure 1H). No differences of intracellular ATP contents were observed (new Figure 1I). We conclude that alterations of extracellular ATP levels are not associated with depletion of intracellular ATP, since the fraction of ATP released from LPS-stimulated WT peritoneal macrophages represent only between 4 to 6% of the total (intra- and extracellular ATP content (new Figure 1—figure supplement 1E). Thus, a slight elevation of intracellular ATP synthesis most likely immediately compensates for the released ATP.

2) ATP released in the extracellular space was shown to be promptly degraded by ecto-nucleotidases, as higher levels of extracellular ATP were observed in the presence of ecto-ATPase inhibitor ARL67156 (Figure 1—figure supplement 1C). Along with NTPDase1/CD39-positive macrophages, other ecto-nucleotidases contribute to the metabolism of ATP in the tissues. For instance, in the liver, NTPDase2 and NTPDase8 are reportedly expressed on portal fibroblasts and bile canaliculi (Fausther et al., 2012; Feldbrügge et al., 2018). Moreover, the presence of soluble ATP-inactivating enzymes freely circulating in the bloodstream (e.g. nucleotide pyrophosphatase/ phosphodiesterase-1 (NPP1): Yegutkin et al., 2003) may lead to the underestimation of actual ATP levels measured in the systemic circulation (and probably, in the peritoneal cavity as well). These alternative and auxiliary inactivating pathways should be mentioned in the text discussed accordingly.

We now discuss this important aspect in the Discussion (fourth paragraph), where we describe the impact of different ecto-ATPases on the systemic and local levels of ATP. This information further strengthens our observation given the possibility of underestimation of measured ATP.

3) Data on markedly down-regulated CD39 mRNA levels in Gap27-treated and CX43-KO LPS-stimulated macrophages (Figure 4F) are interesting but somehow surprising. What might be the mechanism underlying this decrease: does it mean that CD39 expression is regulated by extracellular concentrations of the available ATP substrate? Please, amend or clarify. These data should also be ascertained using direct bioluminescent enzymatic assay and measuring the rate of exogenous ATP decay in control versus treated cells.

As suggested by the reviewers, we performed a functional assay to show extracellular ATP decay in WT compared to MAC-CX43 KO peritoneal macrophages (Figure 4—figure supplement 3C-D). First, we stimulated peritoneal macrophages in vitro for 3 and 6 hours using LPS to induce CD39 expression. Second, we exposed cells to a solution of 100 µM ATP and quantified the bioluminescent enzymatic signal of a luciferin-luciferase reaction proportional to ATP concentration. We observed a decrease of ATP concentration with time in WT and MAC-CX43 KO peritoneal macrophages (Figure 4—figure supplement 3C-D). The kinetics of ATP hydrolysis was not different between WT and MAC-CX43 KO peritoneal macrophages (Figure 4—figure supplement 3C-D). Thus, the difference of CD39 mRNA levels between WT and MAC-CX43 KO peritoneal macrophages seems not to have an impact on extracellular ATP degradation from these cells. The change of expression is potentially the result of autocrine activation in response to elevated levels of pro-inflammatory cytokines secreted by WT macrophages (Deaglio and Robson, 2011). Either, such altered expression does not result in decreased functional CD39 on the cell surface or other ecto-nucleotidases compensate for the decrease of CD39 in the context of CX43 inhibition.

4) In the second paragraph on ATP release via CX43 and MyD88/TRIF, the conclusion is that CX43 is responsible for ATP-release via MyD88/TRIF. However, although the authors show that CX43 expression is lower in MyD88/TRIF KO mice, they do not show that ATP release is downregulated in these cells, and therefore one cannot draw this conclusion based on the available data. Did the authors include an experiment in which ATP release of MyD88/TRIF KO cells was measured, thereby linking the two observations?

As suggested by the reviewer, we now performed an experiment to test LPS-induced ATP release from peritoneal macrophages isolated from MyD88/TRIF KO and WT mice. MyD88/TRIF KO mice and the WT controls were colonized with SPF flora as indicated in the Materials and methods (subsection “Experimental animals”). We observed LPS-induced ATP release from WT but not from MyD88/TRIF KO peritoneal macrophages (Figure 2H). These results support our hypothesis that expression and function of CX43 depends on MyD88/TRIF mediating ATP release in response to LPS.

5) Figure 2F shows that CX43 protein expression in primary murine macrophages is induced by LPS. Perhaps, higher quality/resolution/magnification images would be helpful for better visualization of CX43 staining pattern and subcellular localization in the treated macrophages. Along with the merged image, the authors should provide single channel stainings for Texas Red-labelled CX43 and FITC-labelled macrophages, maybe by showing the representative images only for control cells and those stimulated with LPS for 5 hours.

We now included images at a higher magnification in the supplemental data to allow a better subcellular localisation of CX43 protein in macrophages (new Figure 2—figure supplement 1A). After transcription in the nuclei and translation in the endoplasmic reticulum (ER), CX43 is inserted in the ER membrane, where protein folding and post-translational modifications occur (Epifantseva and Shaw, 2018). After transition through the Golgi network, CX43 proteins are delivered to their target location including the plasma membrane. Importantly, CX43 proteins have a short half-life and are rapidly synthetized and delivered to cellular subdomains (Epifantseva and Shaw, 2018). This explains the strong CX43 expression in the ER and the Golgi.

Single channel stainings for Texas Red-labelled CX43 and FITC-labelled F4/80 were at the indicated time points are now included (new Figure 2—figure supplement 1B).

6) It is unclear to the reviewers how all the ex-vivo/in-vitro experiments with mouse peritoneal macrophages were conducted; it is not clarified in the figure legends or Materials and methods. Is each dot representing 1 mouse? Were all the mouse cells pooled? Why the experiments not conducted in a paired manner using cells from 1 mouse in the different conditions? How many cells were seeded per well?In the same vein, which cells are used is not always specified. What for example does% parents mean in Figure 3F?Such details are important for potential repeating and understanding the figures.

The details about in vitro experiments have now been extended in the Materials and methods (subsection “Isolation and culture of mouse peritoneal macrophages”): Peritoneal macrophages were isolated after pre-treatment using 3% Brewer thioglycollate. Cells from at least 5 different mice were pooled. Pooling was necessary due to the relative low amount of cells obtained per mouse. Thereby, we obtained reasonable amounts of cells to perform the experiments and while reducing the number of animals according to current 3R guidelines. For the same reason, experiments were not conducted in a paired manner. A total number of 40x10^3^ cells were seeded per well of a 96-well plate and 1x10^6^ cells per well of a 6-well plate. For in vitro experiments, a dot is representative of an independent well, as indicated in the figure legends (Figure 1C-H, Figure 4C-E, Figure 4G-H, Figure 1—figure supplement 1C-E, Figure 1—figure supplement 2B, Figure 4 —figure supplement 1A-E, G, Figure 4—figure supplement 2A-C, Figure 4—figure supplement 3A-B). In Figure 4—figure supplement 3C-D, each dot is representative of 5 independent wells. All experiments have been repeated at least three times.

The% of parents in Figure 3F represents percentage of CX43 positive cells for the indicated parent population as detailed in the panel 1, 2 or 3 of Figure 3D. Therefore, the CX43 positive cell fraction of the population identified in 3E is shown.

7) Why were the peritoneal macrophages isolated using thioglycollate? This changes the phenotype of the macrophages tremendously. As it is a potent oxygen scavenger, results should be interpreted with caution if ROS production plays a role. This should be discussed at least in the Discussion.

We agree that peritoneal macrophages isolated from the peritoneal cavity after pre-treatment using 3% Brewer thioglycollate differ from in vivo resident and infiltrating peritoneal macrophages. However, under non-elicited conditions the number of macrophages present in the peritoneal cavity is too low to allow the in vitro experiments we performed. Macrophages isolated from the peritoneal cavity of CLP-operated mice showed high levels of apoptosis in culture and could not be used for in vitro experiments.

Thioglycollate has been extensively used to isolate in a standardized fashion a reasonable amount of macrophages for cell culture and in vitro experiments to assess responses in models of sepsis (Davies et al., 2013; Ghosn et al., 2010). Main differences of thioglycollate-elicited peritoneal macrophages to resident peritoneal macrophages, are the metabolic activity and elevated phagocytic activity (Pavlou et al., 2017). It has also been shown in an old study that thioglycollate-elicited peritoneal macrophages are less efficient than resident macrophages in antibody-dependent cell-mediated cytolysis (Shaw and Griffin, 1982). It cannot be excluded that these functions of macrophages as well as ROS production may have been missed in our experimental setup. However, the main findings we observed were differences in cytokine responses that were observed both, in vivo and in vitro. We now address these points in the Discussion (seventh paragraph), where limitations of our methods are discussed.

8) Looking at Figure 1, in panel C-F, the control group (which should be the same experiment all the time) shows a mean ATP release of 5, 10, 16 and 10 nM respectively. Can the authors explain this? Again, the difference could not be understood from the methods, as they were lacking.

Each experiment with primary cells was always performed at once and in parallel. The different figures represent individual sets of experiments that were repeated at least three times (Figure 1C-G more than 5 times). The differences in-between different sets of experiments may be explained by the extensive protocol that includes multiple steps. This may result in potential differences of viability and consequential altered functionality of cells that then add up to these experiment inherent variabilities we have observed. All efforts to standardize such as using a haematocytometer, controlling of pipetting and temperatures were undertaken to eliminate such bias. Furthermore, ATP release is very sensitive to any stimulus and may be elicited via other mechanisms such as vesicular exocytosis that may additionally explain the differences. Importantly, the differences between experimental conditions remained within each experiment despite such differences in baseline levels.

9) In Figure 1E and F and also Figure 4G and H: solvent controls are missing. Some of the used inhibitors are dissolved in water and do not need a solvent control, but many are not dissolvable in water (such as 18-alpha-glycyrrhetinic acid (chloroform or DMSO), carbenoxolone (in NaOH), P2X4, P2X7 and P2Y2-inhibitors (in DMSO)). How were the inhibitors dissolved and did the authors include solvent controls?

DMSO was used to dissolve water-insoluble 18-alpha-glycyrrhetinic acid and P2 receptors inhibitors. Carbenoxolone was dissolved in distilled water according to manufacturer’s recommendations (Sigma-Aldrich, # C4790). In order to control the impact of DMSO on ATP release, we repeated the experiments using with and without DMSO in WT peritoneal macrophages (new Figure 1—figure supplement 1D). Therefore, peritoneal macrophages were isolated from the peritoneal cavity after thioglycollate pre-treatment as previously described, and exposed to LPS to induce ATP release. In this experiment, DMSO was used at a higher concentration (1%) than the concentration used to dissolve our inhibitors (0.25% for 18-alpha-glycyrrhetinic acid, 0.2% for the purinergic receptors antagonists 5-BDBD, A804598, AR-C 118925XX and MRS2578). We now described precisely solvent used in the Materials and methods (subsection “Compounds”).

10) In Figure 3A, the authors state to have used the same controls for the two western blots. The control blots look very different to me however. Can the authors comment on this? Furthermore, the authors state that CX43 was expressed on cholangiocytes, but the data are missing. Also, they mention Figure 3F twice, stating it shows different things. Is there a figure panel missing? Finally, again in this paragraph, they draw a conclusion which cannot be made on the available data. CX43 is indeed expressed in the liver and lungs as remote target organs, as they show. However, there is no data showing that CX43 is not expressed in other organs. Therefore, the conclusion that CX43 is specifically expressed in these organs is misleading.

- Controls for the two western blot: We apologize to the reviewer for the lack of clarity in the presentation of our western blot results. We include in our resubmission a source data file where the reviewer can refer to the full uncut gels. On the gels we had separate controls, these were however, both quantified and included in the histogram on the right side of Figure 3A. Thus there are 6 controls quantified in the histogram and 6 for each time point. We changed the legend of Figure 3A to describe that controls are different for each experiment but plotted together in the histogram.

- Expression on cholangiocytes: Hepatocytes do not express CX43 under normal conditions (Figure 3B and Figure 3—figure supplement 1A). In our immunohistochemistry staining for CX43 in the liver, we observed CX43 expression on the apical surface of cholangiocytes in livers from control and CLP mice (Figure 3B and Figure 3—figure supplement 1A). This finding is in line with previous data where CX43 has been shown to be expressed on cholangiocytes playing a role in the modulation of bile secretion (Bode et al., 2002; Nathanson et al., 1999). In addition, CX43 is expressed on Kupffer cells, stellate cells and sinusoidal endothelial cells (Maes et al., 2015).

- Figure 3F is showing results from the liver and not from the lungs. Results in the lungs are presented in Figure 3G and H. The text was corrected accordingly (Results, subsection “Connexin-43 expressing macrophages are recruited systemically during peritonitis”.

- CX43 expression is increased in the liver and lungs upon CLP. It is correct that we do not show results from other organs in which it might also be increased during sepsis. By specifically, we alluded to CLP. This sentence has now been adjusted accordingly (Results, subsection “Connexin-43-mediated local and systemic ATP release contributes to macrophage over-activation via P2Y1”).

11) Where the authors studied the expression of pro-inflammatory cytokines, they state some results, which the figures do not confirm. For example, the authors state that Gap27 decreases pro-inflammatory cytokine levels TNF-alpha and IL-6, but only IL-6 is decreased, whereas TNF-alpha is not. The same holds true for ATPgammaS reverting the CX43 blockade: TNF-alpha is not downregulated upon the knockout or Gap27 inhibition, and ATPgammaS does not revert this phenotype, since it is simply absent. Finally, in the assessment of systemic cytokines, again the authors state that IL-6 is decreased, but the data do not support this conclusion.

- The impact of CX43 blocking using Gap27 or CX43 deletion on the secretion of pro-inflammatory cytokines from peritoneal macrophages in vitro is presented in Figures 4C and 4D. CX43 deletion decreased secretion of TNF alpha after 6 hours (Figure 4C) and IL-6 after 3 hours (Figure 4D) of LPS stimulation compared to WT peritoneal macrophages. These findings are now described more precisely in the text (Results, subsection “Connexin-43-mediated local and systemic ATP release contributes to macrophage over-activation via P2Y1”).

- We agree with the observations of the reviewers. Only the secretion of IL-6 is altered in response to ATPgammaS. The text was adjusted accordingly (Results, subsection “Connexin-43-mediated local and systemic ATP release contributes to macrophage over-activation via P2Y1”). Also the interpretation of the results of Figure 5E (in vivo levels of IL-6) were adjusted according to reviewer’s observations (Results, subsection “Improved survival and decreased local and systemic cytokine secretion in response to Connexin-43 blocking or deletion during abdominal sepsis”).

12) How CX43 expression enhances TNF and IL-6 production does not become clear. Is this transcriptional or translational? In addition, isn't the IL-6 not just secondary to TNF? This should at least be discussed if not further supported by experimental data.

It has been shown that extracellular ATP exhibit pleiotropic effects via different purinergic receptors finally leading to altered TNF-alpha and IL-6 signaling in macrophages (Cauwels et al., 2014; Sakaki et al., 2013). The function and downstream signaling of P2Y1 has been well described in platelets or neurons but not in macrophages (Hechler and Gachet, 2015). P2Y1 purinergic receptors are G_αq_-protein coupled receptors that activates phospholipase Cbeta leading to increased cytosolic calcium levels (Barańska et al., 2017). Thus, downstream signaling of P2Y1 mediated TNF-alpha and IL-6 secretion in macrophages is potentially also dependent on these mechanisms (Hunter and Jones, 2015; Jones and Jenkins, 2018). A parallel induction of TNF-alpha and IL-6 transcription following PRRs-dependent TLRs activation given the different transcription factors that regulate IL-6, including NFkappaB, AP-1, SP1, NF-IL-6 and IRF1. As the IL-6 promoter is activated via stimulation by different signals including TNF-alpha it is possible that differences observed in IL-6 levels are a downstream consequence of TNF alpha regulation. This is now mentioned in the Discussion (third paragraph).

13) In line with the previous comment: there is quite some literature (especially Anna Rubartelli's work) showing the role of ATP in the release of IL-1beta out of the macrophage. Since IL-1beta is another deleterious cytokine acting in concert with TNF, but only partially secondary to TNF production, it is a missed chance that the investigators did not include IL-1beta measurements and include the dynamics of that cytokine.

We thank the reviewers for their comment and we included our IL1-beta results in the supplemental data (Figure 5—figure supplement 3C, Results, subsection “Improved survival and decreased local and systemic cytokine secretion in response to Connexin-43 blocking or deletion during abdominal sepsis”). We observed increased IL1-beta levels in the serum of CLP-operated mice compared to controls, but no difference between WT and MAC-CX43 KO mice.

14) What is missed in the Discussion is a more crisp description on the sequence of events, starting with the entry of bacteria in the sterile peritoneal cavity.

We address this point, commenting on the storyline of the paper.

- In order to improve the clarity of the storyline and to guide the readers through the text, we included a graphical Abstract for the re-submission.

- We also include a clear description of the sequence of events in the Discussion part.

- Figure supplements are numbered according to their order of appearance in the text. Because supplemental figures outnumber main figures the numbers do not correspond.

15) The description of the mononuclear phagocytes is confusing: infiltrating monocytes, monocyte-derived macrophages, monocytes, small macrophages, resident peritoneal macrophages, primary peritoneal macrophages.Conceptually, when a bacterial challenge in the peritoneal cavity occurs is that resident macrophages do the first job. Directly followed by the rapid neutrophil response and the slower influx of monocytes (which become exudate macrophages). The term primary macrophages for thioglycollate-induced exudate macrophages is confusing.

In the manuscript, macrophages isolated from the peritoneal cavity without stimulation are described as large and small peritoneal macrophages (LPM and SPM) according to the literature (Ghosn et al., 2010; Wang and Kubes, 2016). LPM are resident macrophages present in the peritoneal cavity and inducing the first response to pathogens (Wang and Kubes, 2016). We observed that in our model of peritonitis, resident LPM are quickly replaced by blood-derived infiltrating SPM which is supported by previous observations (Wang and Kubes, 2016). In addition to primary LPM and SPM, thioglycollate elicited peritoneal macrophages were isolated for experimental purposes. The description of macrophages in the manuscript was standardized as follows: (1) Large peritoneal macrophages (LPM) represent resident peritoneal macrophages (2) Small peritoneal macrophages (SPM) represent infiltrating macrophages and (3) Thioglycollate elicited peritoneal macrophages are termed peritoneal macrophages throughout the manuscript. Other descriptions including infiltrating monocytes, monocyte-derived macrophages, monocytes and primary peritoneal macrophages are omitted.